# GENEXAM: A MULTIDISCIPLINARY TEXT-TO-IMAGE EXAM

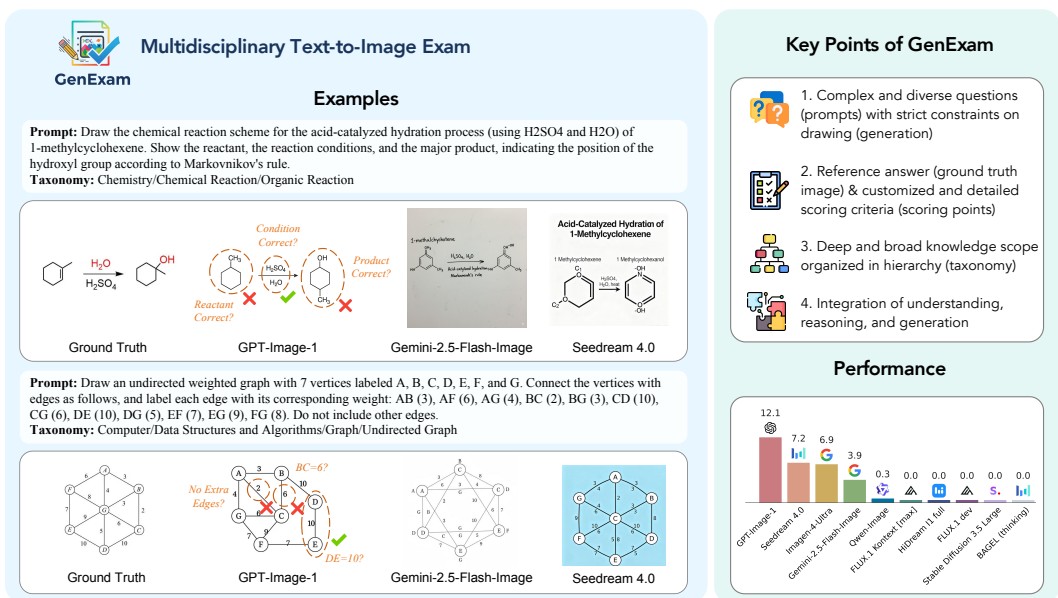

Figure 1: **Examples and performance of state-of-the-art text-to-image models on GenExam.** Orange dashed circles indicate scoring points. GenExam contains complex and diverse prompts resembling human exams, and pose great challenge to existing models.

## ABSTRACT

Exams are a fundamental test of expert-level intelligence and require integrated understanding, reasoning, and generation. Existing exam-style benchmarks mainly focus on understanding and reasoning tasks, and current generation benchmarks emphasize the illustration of world knowledge and visual concepts, neglecting the evaluation of rigorous drawing exams. We introduce GenExam, the first benchmark for multidisciplinary text-to-image exams, featuring 1,000 samples across 10 subjects with exam-style prompts organized under a four-level taxonomy. Each problem is equipped with ground-truth images and fine-grained scoring points to enable a precise evaluation of semantic correctness and visual plausibility. Experiments show that even state-of-the-art models such as GPT-Image-1 and Gemini-2.5-Flash-Image achieve less than 15% strict scores, and most models yield almost 0%, suggesting the great challenge of GenExam. By framing image generation as an exam, GenExam offers a rigorous assessment of models' ability to integrate understanding, reasoning, and generation, providing insights on the path to general AGI. Our benchmark and evaluation code will be released.

## 1 INTRODUCTION

Exams are the ultimate test of expert-level intelligence. They are not merely about recalling knowledge points, but serve as a comprehensive assessment of understanding, reasoning, and generation.

The ability to solve multidisciplinary exam problems indicates that a model has expert-level intelligence surpassing that of most adults. Therefore, exam-style benchmarks naturally serve as a crucial yardstick to evaluate the progress of artificial general intelligence (AGI) towards domain expertise (Morris et al., 2023; Yue et al., 2024a; Phan et al., 2025).

However, existing multidisciplinary benchmarks are primarily focused on understanding tasks, including pure text benchmarks (Wang et al., 2024b; Zhong et al., 2023) and multimodal benchmarks (Yue et al., 2024a;b), but rarely focus on image generation tasks. In the field of text-to-image generation (T2I), current benchmarks focus more on general world knowledge reasoning (Niu et al., 2025; Sun et al., 2025). Some benchmarks have explored the domain of subject-specific images but are largely limited to concept illustration (Luo et al., 2025; Chang et al., 2025b;a), a relatively simple task with loose diagnostic criteria, similar to "illustrating concepts through image generation" rather than "solving a drawing exam".

Upon a closer inspection of graph-drawing questions in common multidisciplinary exams, such as AP (CollegeBoard, 1952), A-level (Cambridge, 1951), and IB (IBO, 1968), we find that text-to-image exams present far distinct challenges against common image generation tasks:

1. The questions (prompts) are typically more complex, precise, and diverse, with strict and explicit constraints on drawing (generation);
2. Each question is equipped with a reference answer (ground truth image) and detailed scoring criteria (scoring points), enabling rigorous evaluation of the correctness of drawn images;
3. Knowledge scope is broader and deeper and can be systematically organized into a hierarchical structure (taxonomy);
4. Solving them demands rigorous semantic accuracy in drawn images, requiring integration of *understanding, reasoning, and generation*.

As shown in Fig. 1, even state-of-the-art text-to-image models, such as GPT-Image-1 (OpenAI, 2025b) and Gemini-2.5-Flash-Image (Nano Banana) (Google, 2025), excel at producing images with superficially plausible overall structure but often fall short in portraying correct exam details.

Based on this motivation, we introduce *GenExam*, the first benchmark dedicated to multidisciplinary text-to-image exams. As shown in Fig. 2, GenExam asks the model to accomplish the subject-specific drawing exam with a series of detailed, rigorous and knowledgeable requirements. It includes 1,000 samples of 10 subjects with detailed four-level taxonomy, collected from college-level exams and open-source benchmarks and carefully reviewed by PhD annotators. In addition, GenExam adopts a strict prompt generation process through cooperation of GPT-5 (OpenAI, 2025a) and humans, with each prompt designed in the style of real exams (CollegeBoard, 1952; IBO, 1968). Given these designs, models are required to seamlessly integrate their understanding, reasoning and generation capabilities to solve the problem of GenExam.

The evaluation of multidisciplinary images also poses challenges, as they prioritize semantic correctness over photorealism or aesthetic quality, different from natural images. Relying solely on a single instruction for MLLM-as-a-judge evaluation (Zhang et al., 2025b) makes it difficult to cover all requirements in the prompt. Therefore, we design customized evaluation criteria for each individual sample, similar to the exam marking process. As shown in in Fig. 1, scoring points of each sample are produced through a rigorous review process of GPT-5 and humans, with the ground truth image as the evaluation reference. Leveraging the most advanced commercial MLLM, *i.e.* GPT-5, we can automatically diagnose each scoring point through visual question answering. In addition to the correctness evaluation, GenExam also includes additional metrics to assess the visual plausibility, namely encompassing spelling, readability, and logical consistency. Based on the above metrics, we calculate two final scores to satisfy different evaluation requirements, with strict and relaxed standards respectively.

Our experimental results show that even state-of-the-art text-to-image models such as GPT-Image-1 (OpenAI, 2025b) and Gemini-2.5-Flash-Image (Google, 2025) can only achieve strict scores of less than 15%. Many of the latest T2I and unified multimodal large language models (MLLMs) like FLUX.1 Kontext max (Batifol et al., 2025), Qwen-Image (Wu et al., 2025) and BAGEL (Deng et al., 2025) even yield almost 0% strict scores, highlighting the great challenge of GenExam. Our experiments also indicate a significant gap between open-source and closed-source models. All these empirical studies suggest a long path ahead for existing models towards general AGI. Overall, our contributions can be summarized in three folds:

- We propose GenExam, the first benchmark for multidisciplinary text-to-image exams, with carefully curated prompts and ground truth images. It aims to serve as a real exam to test whether models can seamlessly integrate understanding, reasoning and generation.
- We design an evaluation framework tailored for multidisciplinary images, assessing the semantic correctness and visual plausibility of disciplinary images through scoring points customized for each sample.
- Comprehensive experiments on existing T2I models and unified MLLMs highlight the challenge of GenExam. We provide analysis and insights on their capabilities and areas for improvement.

## 2 RELATED WORK

### 2.1 TEXT-TO-IMAGE GENERATION MODELS

Text-to-image generation (T2I) has been a primary research focus in generative AI in recent years. The dominant approaches are based on diffusion (Ho et al., 2020; Dhariwal & Nichol, 2021; Rombach et al., 2022; Labs, 2024) or autoregressive objects (Ramesh et al., 2021; Tian et al., 2024; Sun et al., 2024), which excel at generating images with high aesthetic quality and strong semantic consistency with prompts. Recently, unified MLLMs have emerged as a popular research topic, which supports multimodal understanding and image generation with a single model (Wang et al., 2024a; Deng et al., 2025; Xie et al., 2025; Li et al., 2025). The latest state-of-the-art models like GPT-Image-1 (OpenAI, 2025b) and Gemini-2.5-Flash-Image (Google, 2025) show strong capabilities in generating plausible natural and aesthetic images. However, as shown in our experiments, their ability on multidisciplinary text-to-image exams is still limited.

### 2.2 TEXT-TO-IMAGE GENERATION BENCHMARKS

Early evaluation of text-to-image generation is mainly based on image similarity metrics, *e.g.* FID (Heusel et al., 2017). These methods often fail to capture the higher-level semantics of the generated images. Current evaluation for T2I focuses primarily on literal prompt-image alignment, with benchmarks like GenEval (Ghosh et al., 2023) and metrics like CLIP score (Hessel et al., 2021) and VQA score (Lin et al., 2024). Recent works move towards reasoning-informed generation (Niu et al., 2025; Zhao et al., 2025; Meng et al., 2024; Sun et al., 2025; Zhang et al., 2025a), and typically rely on MLLM-as-a-judge (Zhang et al., 2025b) for evaluation. However, their focus is mainly on world knowledge or commonsense reasoning, with domains restricted to natural or synthetic images. Meanwhile, benchmarks such as MMMG (Luo et al., 2025), OneIG-Bench (Chang et al., 2025a), and SridBench (Chang et al., 2025b) explore multidisciplinary image generation, but primarily in the form of knowledge-concept illustration, a simple task with loose diagnostic criteria compared to exam-style questions. Existing benchmarks for multidisciplinary exams are mainly for understanding tasks (Wang et al., 2024b; Yue et al., 2024a; Xia et al., 2024; Phan et al., 2025), but rarely focus on image generation. In contrast, our GenExam targets multidisciplinary text-to-image exams, whose prompts are complex, precise, and diverse and pose strict and explicit constraints on generation. By designing fine-grained scoring points, GenExam provides a more rigorous and accurate evaluation of reasoning-informed generation in multidisciplinary images.

## 3 GENEXAM

### 3.1 OVERVIEW

Fig. 2 provides an overview of our GenExam. It contains 1,000 samples that span over 10 subjects: mathematics, physics, chemistry, biology, computer science, geography, economics, music, history, and engineering. Each sample contains a ground truth image, a four-level taxonomy, and several scoring points to judge the semantic correctness of the image.

**Scoring Points.** One critical challenge in the evaluation of exam-style image generation is that it is difficult to judge the semantic correctness of the generated image. This becomes evident when using MLLM-as-a-judge (Zhang et al., 2025b) with a single instruction template, where MLLMs often fail to cover all the requirements specified in the prompt. Examples include the specific structure of organic molecules (*e.g.*, the number of each type of atom, functional groups, chemical bonds),

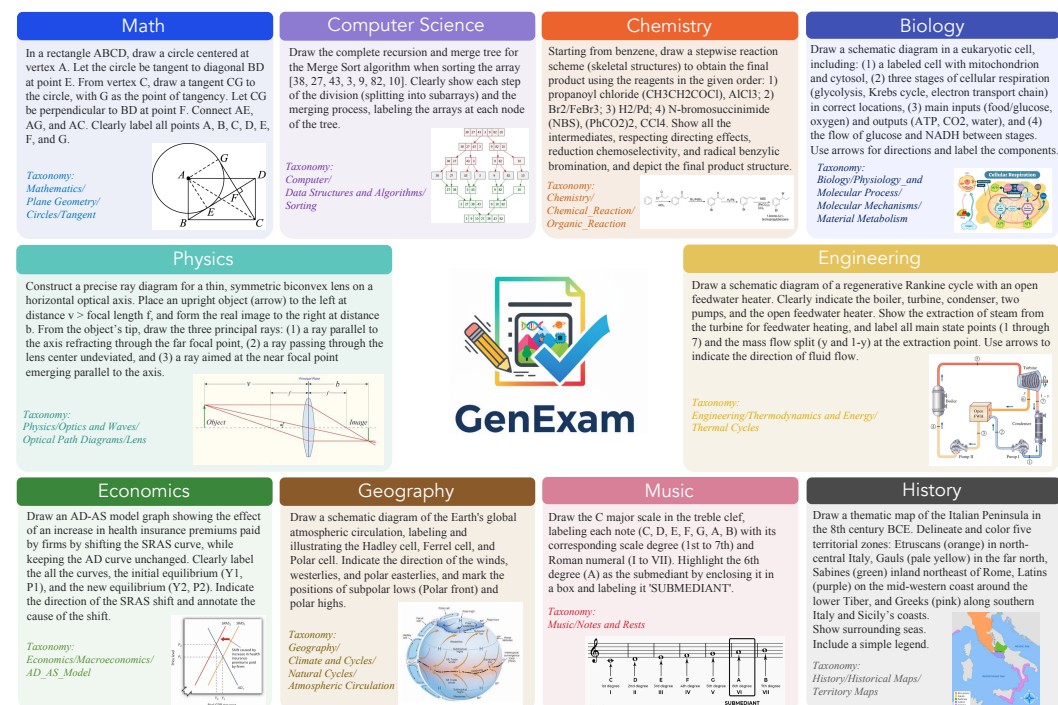

Figure 2: **Examples from GenExam.** GenExam contains 1,000 exam-style prompts that span over 10 subjects and corresponding reference images for multidisciplinary text-to-image exam.

positional relationships in geometric figures, all notes in a musical score, etc. Therefore, inspired by scoring criteria for graph-drawing questions in human exams, we design several scoring points for each prompt, in the form of questions. By answering the questions, we can decide whether the generated image is correct, *e.g.* "Does the molecule contain exactly 8 carbon atoms?". This ensures that each sample has a unique evaluation criterion, which improves the accuracy and stability of the evaluation. Each scoring point has a predefined score that sums up to 1, similar to the grading system in exams. The detailed evaluation protocol is given in Sec. 3.3.

**Taxonomy.** In addition to the top-level subject, we also provide a four-level taxonomy, such as "Mathematics/Plane Geometry/Circles/Tangent" in Fig. 2. The taxonomy annotations are built based on ISCED-F (UNESCO, 2013) fields, with the aim of improving the academic strictness of GenExam for seamless integration with multidisciplinary research. Detailed four-level taxonomy is provided in Appendix D.

**Statistics.** In Tab. 1, the prompts in GenExam exhibit an average length of 74.8 words, providing dense and detailed requirements for the generated images, resembling difficult graph-drawing questions in real human exams. Each prompt has an average of 6.9 scoring points to thoroughly assess the generated image from all aspects based on the input prompt. In Fig. 3, we visualize the distributions of the image types, which demonstrate the diversity of images in our benchmark. We also provide a subset of 250 images to facilitate efficient and more affordable evaluation. More statistics are provided in Appendix A.

## 3.2 DATA CURATION PIPELINE

**Data Collection.** The data curation pipeline is shown in Fig. 4. To construct a general benchmark for multidisciplinary text-to-image exam, we first consider possible subjects and domains of multidisciplinary images and curate a four-level taxonomy assisted by GPT-5 (OpenAI, 2025a) and through manual check. We then use these taxonomy to generate keywords related to specific subject knowledge, and collect web images from exams and textbooks based on Google search. Images from existing subject-knowledge-related datasets are also collected: MMMU (Yue et al., 2024a),

| Statistics | Value |
|---|---|
| Taxonomy Count (level 1/2/3/4) | 10/40/132/236 |
| Subject Knowledge Difficulty (easy:medium:hard) | 24%: 38%: 38% |
| Minimum Number of Scoring Points | 3 |
| Maximum Number of Scoring Points | 14 |
| Average Number of Scoring Points | 6.9 |
| Minimum Prompt Length (words) | 24 |
| Maximum Prompt Length (words) | 173 |
| Average Prompt Length (words) | 74.8 |

Table 1: **Key statistics of GenExam.**

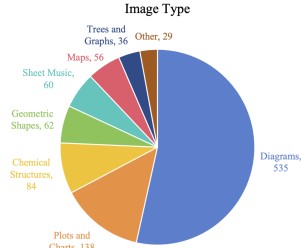

Figure 3: **Distribution of image types.**

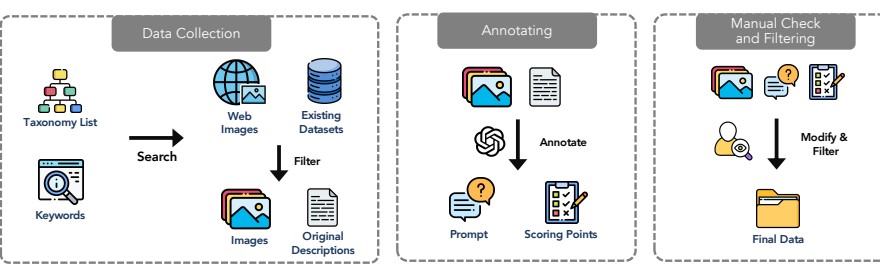

Figure 4: **Data curation pipeline.** We use pre-defined taxonomy to collect web images and existing datasets, and conduct annotating and filtering based on GPT-5 and manual check.

ScienceQA (Lu et al., 2022), TextbookQA (Kembhavi et al., 2017), MAVIS (Zhang et al., 2024), ChEBI (Hastings et al., 2016), HLE (Phan et al., 2025) and R-Bench (Guo et al., 2025). We obtain 40K images from the data sources with their original text descriptions (*e.g.* webpage content for web images, lecture notes in TextbookQA, or question and answers in MMMU). We then conduct automatic filtering by utilizing GPT-5 to rate images in terms of text richness, image domain, complexity and subject knowledge density, and set a threshold to remove undesirable images, obtaining 6.5K images. Details of automatic filtering and proportion of image sources are in Appendix B.

**Annotating.** We employ GPT-5 to generate the initial version of prompts and scoring points. Prompts that require subject knowledge and reasoning are generated, similar to graph-drawing questions in exams. The original text description from the data source is used for reference to improve the precision, bu should be treated carefully since they may contain irrelevant information. Subsequently, scoring points (questions and scores) are generated based on the prompt. We enforce through prompt engineering that the questions cannot be too general (*e.g.* "Does the image show the correct structure of benzene?") or too complex (can be split into multiple sub-questions). Scores are based on the difficulty of the questions, where basic questions should have low scores and harder ones have high scores. We provide some few-shot examples to enhance the labeling process. The complete prompts are provided in Appendix F.2.

**Manual Check and Filtering.** Three PhD annotators perform rigorous manual inspection on all the images, prompt, and scoring points, and cross-validate each other's decision. Images with duplicate domain or undesirable text richness and complexity not recognized by automatic filtering are first manually removed. Subsequently, imprecise prompts and are manually modified to align with the ground truth images and include sufficient disciplinary knowledge. Inaccurate scoring points are removed or updated to match the prompt. We finally obtain 1K images as the final dataset.

### 3.3 EVALUATION PROTOCOL

Unlike the evaluation of natural images, multidisciplinary image generation focuses more on correctness than photorealism or aesthetic quality, and a simple flaw can cause significant errors. For instance, if a single atom is drawn mistakenly in a chemical structure or some arrows are depicted in the opposite direction in a diagram, the entire image is incorrect. Similar examples include arrows pointing at the opposite direction, wrongly labeled points, etc. In practice, common failure cases of current T2I models (Fig. 7) are either from semantic consistency with the prompt or plausibility of the image itself (*e.g.* spelling errors). Therefore, to thoroughly judge the correctness and over-

Figure 5: **Evaluation protocol of GenExam.** Employing MLLM-as-a-judge, we use scoring points of each prompt to calculate semantic correctness (0-1), and also rate the image in terms of spelling, logical consistency, and readability (0/1/2). The four dimensions are used to calculate a strict score and a relaxed score.

all quality of the generated image, we consider two dimensions: semantic correctness and visual plausibility. The evaluation pipeline is shown in Fig. 5.

**Semantic Correctness.** This dimension assesses the image's consistency with the input prompt. The scoring points mentioned in Sec. 3.1 provide detailed criteria to judge each specific prompt. We use the MLLM judge to answer each scoring points questions by "Yes" or "No" like visual question answering. Since the answers should all be positive for a correct image, we can compute semantic correctness score as the sum of scores of all the questions whose answers are "Yes". The range of semantic correctness is 0-1. Since judging certain subject knowledge-related aspects can be difficult, we add the ground truth image as input, so that the MLLM judge can use it as reference.

**Visual Plausibility.** This dimension focuses on the image itself, consisting of three sub-dimensions:

1. **Spelling**: Whether the spelling of the text in the image is correct, including notation and equations. This assesses the model's text rendering capability.
2. **Logical Consistency**: Whether all marks, text, musical notes, etc. are logically consistent, *e.g.* the labeled coordinates and the actual position. This evaluates the model's ability to maintain internal coherence in the generated content, reflecting its capacity to integrate multiple components in a logically unified manner.
3. **Readability**: Whether all components are clearly readable and identifiable, without incorrectly placed labels and marks, overlap, occlusion, and unlabeled key components. This requires the model to design an appropriate layout and place the components, potentially through reasoning.

All the three sub-dimensions are in range 0-2, where 2 indicates almost perfect (tiny errors are allowed), 1 indicates a few flaws that slightly hinder the understanding of the key information, and 0 indicates critical errors that significantly hinder the understanding of the key information.

**Strict and Relaxed Scores.** Based on the two dimensions, we first calculate a strict score as the percentage of correct images among all generated images. An image is considered correct if and only if semantic correctness is 1 (*i.e.* all scoring points are correct) and the scores of spelling, logical consistency and readability are all 2, indicating full satisfaction of all evaluation dimensions, similar to RISEBench (Zhao et al., 2025). It is only more aligned with our intention for multidisciplinary image generation where a tiny flaw could lead to a technically wrong image, posing a great challenge for T2I models.

However, as shown in Tab. 2, existing methods struggle to generate strictly correct images on GenExam. To highlight the difference between these models, we additionally calculate a relaxed score as the weighted average of the semantic correctness, spelling, logical consistency and readability. The weights are selected based on alignment with human preferences, similar to the process in WiScore (Niu et al., 2025), and the values are 0.7, 0.1, 0.1 and 0.1 respectively. We encourage future research to report both strict and relaxed scores for comparison.

**Evaluator Model.** We utilize MLLM-as-a-judge (Zhang et al., 2025b) for the above evaluation with carefully crafted prompts, provided in Appendix F.3. Specifically, we adopt GPT-5 (OpenAI,

| Model | Math | | Phy | | Chem | | Bio | | Geo | | Comp | | Eng | | Econ | | Music | | Hist | | Overall | |
|---|---|---|---|---|---|---|---|---|---|---|---|---|---|---|---|---|---|---|---|---|---|---|
| | Str | Rel | Str | Rel | Str | Rel | Str | Rel | Str | Rel | Str | Rel | Str | Rel | Str | Rel | Str | Rel | Str | Rel | Str | Rel |
| *Closed-source Models* | | | | | | | | | | | | | | | | | | | | | | |
| GPT-Image-1 | **8.0** | **52.0** | **13.2** | **66.4** | **13.5** | **53.4** | **22.8** | **74.6** | **15.9** | **73.9** | **10.3** | **55.6** | **13.1** | 65.5 | **13.0** | **65.8** | **9.3** | **52.6** | 2.4 | **67.4** | **12.1** | **62.6** |
| Seedream 4.0 | 2.6 | 39.8 | 3.5 | 49.0 | 5.9 | 46.1 | 18.6 | 71.0 | 10.6 | 65.1 | 6.9 | 52.2 | 11.7 | 60.0 | 5.2 | 56.0 | 0.0 | 34.5 | **7.3** | 56.7 | 7.2 | 53.0 |
| Imagen-4-Ultra | 2.6 | 35.9 | 9.7 | 57.4 | 9.3 | 44.5 | 14.7 | 68.1 | 7.6 | 66.9 | 2.9 | 40.1 | 12.6 | 65.6 | 9.1 | 59.7 | 0.0 | 38.4 | 0.0 | 57.8 | 6.9 | 53.4 |
| Gemini-2.5-Flash-Image | 0.7 | 43.1 | 7.1 | 60.9 | 4.2 | 45.3 | 5.1 | 72.6 | 4.5 | 70.2 | 4.9 | 47.4 | 10.0 | **65.8** | 1.3 | 59.8 | 1.5 | 37.0 | 0.0 | 57.1 | 3.9 | 55.9 |
| Seedream 3.0 | 0.7 | 18.6 | 0.0 | 21.5 | 0.8 | 18.3 | 0.0 | 32.2 | 0.0 | 38.2 | 0.0 | 15.3 | 0.0 | 26.5 | 0.0 | 12.5 | 0.0 | 21.6 | 0.0 | 29.2 | 0.2 | 23.4 |
| FLUX.1 Kontext max | 0.0 | 23.5 | 0.0 | 25.6 | 0.0 | 19.2 | 0.0 | 38.3 | 0.0 | 47.5 | 0.0 | 20.9 | 0.0 | 28.9 | 0.0 | 22.3 | 0.0 | 25.4 | 0.0 | 33.5 | 0.0 | 28.5 |
| *Open-source T2I Models* | | | | | | | | | | | | | | | | | | | | | | |
| Qwen-Image | 0.0 | 18.9 | 0.0 | 26.3 | 0.0 | 15.3 | 0.0 | 32.1 | 3.0 | 49.6 | 0.0 | 18.9 | 0.0 | 32.0 | 0.0 | 20.3 | 0.0 | 23.4 | 0.0 | 38.6 | 0.3 | 27.5 |
| HiDream-I1-Full | 0.0 | 16.7 | 0.0 | 17.7 | 0.0 | 13.5 | 0.0 | 27.3 | 0.0 | 36.2 | 0.0 | 15.4 | 0.0 | 24.4 | 0.0 | 18.8 | 0.0 | 21.3 | 0.0 | 31.8 | 0.0 | 22.3 |
| FLUX.1 dev | 0.0 | 12.2 | 0.0 | 14.4 | 0.0 | 12.5 | 0.0 | 22.8 | 0.0 | 36.4 | 0.0 | 11.0 | 0.0 | 14.0 | 0.0 | 9.2 | 0.0 | 21.3 | 0.0 | 21.7 | 0.0 | 17.6 |
| FLUX.1 Krea | 0.0 | 7.0 | 0.0 | 14.0 | 0.0 | 8.5 | 0.0 | 26.5 | 0.0 | 38.4 | 0.0 | 8.4 | 0.0 | 15.4 | 0.0 | 11.1 | 0.0 | 16.8 | 0.0 | 17.4 | 0.0 | 16.4 |
| Stable Diffusion 3.5 Large | 0.0 | 12.2 | 0.0 | 13.2 | 0.0 | 10.7 | 0.0 | 21.8 | 0.0 | 38.8 | 0.0 | 6.6 | 0.0 | 16.3 | 0.0 | 8.0 | 0.0 | 24.1 | 0.0 | 18.0 | 0.0 | 17.0 |
| *Open-source Unified MLLMs* | | | | | | | | | | | | | | | | | | | | | | |
| BAGEL (thinking) | 0.0 | 11.7 | 0.0 | 13.8 | 0.0 | 11.9 | 0.0 | 15.2 | 0.0 | 28.5 | 0.0 | 6.2 | 0.0 | 10.7 | 0.0 | 6.3 | 0.0 | 14.7 | 0.0 | 16.0 | 0.0 | 13.5 |
| BAGEL | 0.0 | 14.7 | 0.0 | 10.6 | 0.0 | 7.9 | 0.0 | 10.8 | 0.0 | 24.5 | 0.0 | 6.8 | 0.0 | 10.2 | 0.0 | 5.3 | 0.0 | 13.7 | 0.0 | 14.4 | 0.0 | 11.9 |
| Show-o2-7B | 0.0 | 10.8 | 0.0 | 11.9 | 0.0 | 4.8 | 0.0 | 12.8 | 0.0 | 33.3 | 0.0 | 4.7 | 0.0 | 11.8 | 0.0 | 7.0 | 0.0 | 8.8 | 0.0 | 14.5 | 0.0 | 12.0 |
| Show-o2-1.5B-HQ | 0.0 | 7.3 | 0.0 | 7.5 | 0.0 | 6.2 | 0.0 | 15.0 | 0.0 | 25.3 | 0.0 | 4.3 | 0.0 | 9.3 | 0.0 | 7.3 | 0.0 | 7.6 | 0.0 | 19.8 | 0.0 | 11.0 |
| BLIP3o-NEXT-GRPO-Text-3B | 0.0 | 15.5 | 0.0 | 10.5 | 0.0 | 9.2 | 0.0 | 15.5 | 0.0 | 23.7 | 0.0 | 8.2 | 0.0 | 10.1 | 0.0 | 8.1 | 0.0 | 15.2 | 0.0 | 10.2 | 0.0 | 12.6 |
| BLIP3o-8B | 0.0 | 6.4 | 0.0 | 5.5 | 0.0 | 4.7 | 0.0 | 7.0 | 0.0 | 16.7 | 0.0 | 3.6 | 0.0 | 8.4 | 0.0 | 2.5 | 0.0 | 6.0 | 0.0 | 11.2 | 0.0 | 7.2 |
| Janus-Pro | 0.0 | 13.7 | 0.0 | 8.8 | 0.0 | 8.2 | 0.0 | 7.2 | 0.0 | 18.8 | 0.0 | 3.9 | 0.0 | 10.5 | 0.0 | 4.2 | 0.0 | 14.5 | 0.0 | 6.6 | 0.0 | 9.6 |
| Emu3 | 0.0 | 11.3 | 0.0 | 0.6 | 0.0 | 0.6 | 0.0 | 5.6 | 0.0 | 34.6 | 0.0 | 5.1 | 0.0 | 16.5 | 0.0 | 1.9 | 0.0 | 5.8 | 0.0 | 6.2 | 0.0 | 8.8 |

Table 2: **Strict scores (Str) and relaxed scores (Rel) on GenExam.** GPT-Image-1 achieves the highest strict score of 12.1%, and open-source models yield nearly 0%, highlighting the great challenge of multidisciplinary text-to-image exam.

2025a), which shows strong multimodal understanding capabilities in multidisciplinary images. In Appendix Tab. 4, we also compare the robustness of other MLLMs with their API costs and time.

## 4 EXPERIMENTS

In this section, we evaluate the performance of representative methods on our GenExam. We also validate the alignment between MLLM-as-a-judge and human evaluation. We use gpt-5-2025-08-07 (OpenAI, 2025a) as the evaluator model and set the reasoning effort as "low". More experiments are provided in Appendix C, including results categorized by subject knowledge difficulty and level-2 taxonomy, detailed scores of each dimension, and ablations on the evaluator model.

### 4.1 BASELINES

We select 18 representative models for comparison, including closed-source proprietary models, open-source T2I models and unified MLLMs. We use the default T2I configuration for each of the models for inference.

**Closed-source Models:** GPT-Image-1 (GPT-4o) (OpenAI, 2025b), Gemini-2.5-Flash-Image (Nano Banana) (Google, 2025), Imagen-4-Ultra (Deepmind, 2025), Seedream 4.0 (Seed, 2025), Seedream 3.0 (Gao et al., 2025), and FLUX.1 Kontext max (Batifol et al., 2025). These proprietary models represent the state-of-the-art methods in text-to-image generation.

**Open-source T2I Models:** Qwen-Image (Wu et al., 2025), FLUX.1 dev (Labs, 2024), FLUX.1 Krea (Lee et al., 2025), HiDream-I1-Full (Cai et al., 2025), and Stable Diffusion 3.5 Large (Rombach et al., 2022). These models are dedicated to the T2I task.

**Open-source Unified MLLMs:** Show-o2-7B, Show-o2-1.5B-HQ (Xie et al., 2025), BAGEL (Deng et al., 2025) (thinking & non-thinking), Janus-Pro (Chen et al., 2025c), Emu3 (Wang et al., 2024a), BLIP3o-8B (Chen et al., 2025a), and BLIP3o-NEXT-GRPO-Text-3B (Chen et al., 2025b). These models employ a unified architecture for T2I and multimodal understanding.

### 4.2 MAIN RESULTS

Results on GenExam are provided in Tab. 2. For strict scores, we observe that all models struggle in multidisciplinary text-to-image exams, despite their superior capability in general T2I tasks. This

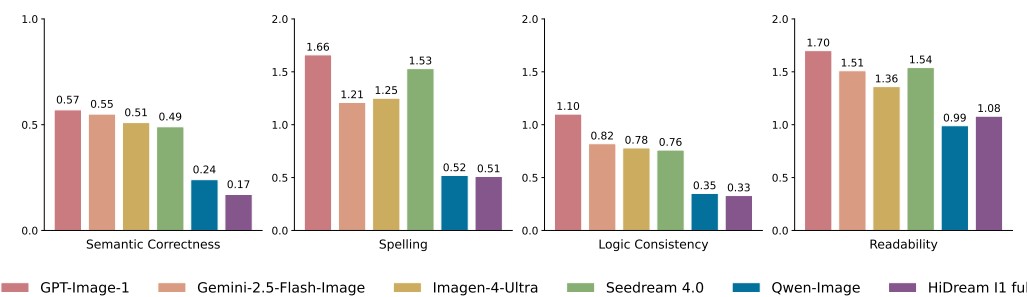

Figure 6: **Comparison across models on four evaluation dimensions.**

|  | Semantic Correctness | Spelling | Logical Consistency | Readability |
|---|---|---|---|---|
| Range | 0-1 | 0-2 | 0-2 | 0-2 |
| MAE | 0.10 | 0.11 | 0.28 | 0.20 |

(a) MAE

| Metric | Kendall | | Spearman | | Pearson | |
|---|---|---|---|---|---|---|
|  | $\tau$ | $p$ | $\rho$ | $p$ | $r$ | $p$ |
| Relaxed Score | **0.6746** | <0.0001 | **0.8217** | <0.0001 | **0.8444** | <0.0001 |
| Semantic Correctness | 0.6333 | <0.0001 | 0.7862 | <0.0001 | 0.8062 | <0.0001 |
| VQA Score | 0.1452 | 0.0009 | 0.2189 | 0.0005 | 0.1786 | 0.0046 |
| CLIP Score | 0.1158 | 0.0072 | 0.1620 | 0.0103 | 0.1647 | 0.0091 |

(b) Correlation

Table 3: **Mean absolute error (MAE) and correlations between human preferences and automatic evaluation.**

demonstrates the great challenge of our benchmark. Specifically, GPT-Image-1 achieves the highest strict score of 12.1%, highlighting its relatively strong ability to integrate multidisciplinary knowledge understanding and reasoning into image generation. Seedream 4.0, Gemini-2.5-Flash-Image and Imagen-4-Ultra are also capable of generating correct images in some cases, with overall strict scores of 7.2%, 6.9% and 3.9%, respectively. Seedream 3.0 and FLUX.1 Kontext max fail in almost all the cases. Open-source models all achieve near-zero strict scores, leaving a huge gap with leading closed-source proprietary models.

The differences between the models are highlighted with relaxed scores. The performance gap between open-source and closed-source models still exists, with GPT-Image-1 being the top-performing method (62.6 relaxed score). Assisted with a multimodal understanding MLLM (Qwen2.5-VL (Bai et al., 2025b)) for feature extraction, Qwen-Image achieves the highest relaxed score of 27.5 among open-source methods, showing the necessity of integrating understanding ability into T2I. However, current unified MLLMs like BAGEL and Show-o2 still lag behind dedicated T2I models, suggesting a significant challenge for unified MLLMs to utilize its multidisciplinary knowledge and reasoning ability into image generation.

To gain deeper insights into the strengths and limitations of the models, we provide a comparison of representative models on each evaluation dimension in Fig. 6. Specifically, GPT-Image-1 surpasses other models across all four dimensions, notably with a significant advantage in spelling and logic consistency. Gemini-2.5-Flash-Image, Imagen-4-Ultra and Seedream 4.0 also exhibit leading performance. For open-source models like Qwen-Image and HiDream-I1-Full, their readability is relatively close to the top models, but the other three dimensions still pose great challenges.

### 4.3 HUMAN ALIGNMENT

We conduct a human alignment experiment to examine the validity of MLLM-as-a-judge, following the paradigms in (Zhao et al., 2025; Sun et al., 2025). Five experts are asked to score 250 randomly sampled images generated by GPT-Image-1. They first provide an overall rating between 1 and 10, and then follow the automatic evaluation protocol to annotate scoring points, spelling, logical consistency and readability. The average scores among all experts are used to calculate the mean absolute error (MAE) and correlations between MLLM-judge predicted scores and human scores. In Tab. 3(a), the results show a low MAE across the four dimensions, indicating high precision of our evaluation pipeline. We then use the overall rating to calculate correlations (Kendall's $\tau$, Spearman's $\rho$ and Pearson's $r$) between human scores and four automatic metrics: our relaxed score, our semantic correctness, VQA score (Lin et al., 2024), and CLIP score (Hessel et al., 2021). In Tab. 3(b), our relaxed score achieves high correlations with human judgments with $p < 0.05$,

| Evaluator Model | Ground Truth Input | Scoring Points Accuracy ↑ | Plausibility MAE ↓ | Kendall $\tau$ ↑ | Spearman $\rho$ ↑ | Pearson $r$ ↑ | API Cost ($/img) | Time (s/img) |
|---|---|---|---|---|---|---|---|---|
| *Closed-source MLLMs* | | | | | | | | |
| GPT-5 + low reasoning effort | Img | **88.5** | **0.20** | **0.6746** | 0.8217 | **0.8444** | 0.0242 | 72 |
| GPT-5 + low reasoning effort | – | 86.2 | 0.23 | 0.6141 | 0.7753 | 0.7976 | 0.0231 | 59 |
| GPT-5 + low reasoning effort | Text | 87.8 | 0.21 | 0.6559 | 0.8103 | 0.8336 | 0.0234 | 63 |
| GPT-5 + low reasoning effort | Img, Text | **88.5** | **0.20** | 0.6693 | **0.8345** | 0.8427 | 0.0250 | 75 |
| GPT-5 + minimal reasoning effort | Img | 84.2 | 0.28 | 0.6066 | 0.7521 | 0.7860 | 0.0144 | 35 |
| GPT-5 + medium reasoning effort | Img | 86.7 | 0.21 | 0.6617 | 0.8224 | 0.8358 | 0.0389 | 101 |
| GPT-5 + high reasoning effort | Img | 87.6 | **0.20** | 0.6647 | 0.8190 | 0.8304 | 0.0654 | 186 |
| GPT-5-mini + minimal reasoning effort | Img | 79.0 | 0.35 | 0.5446 | 0.6996 | 0.7185 | 0.0039 | 11 |
| GPT-5-mini + low reasoning effort | Img | 84.3 | 0.29 | 0.6002 | 0.7729 | 0.7806 | 0.0052 | 19 |
| GPT-5-mini + medium reasoning effort | Img | 84.2 | 0.25 | 0.6313 | 0.8025 | 0.8165 | 0.0079 | 36 |
| GPT-5-mini + high reasoning effort | Img | 83.4 | 0.26 | 0.5906 | 0.7611 | 0.7811 | 0.0196 | 119 |
| o3 | Img | 82.0 | 0.23 | 0.6082 | 0.7706 | 0.7895 | 0.0229 | 39 |
| o4-mini | Img | 84.7 | 0.35 | 0.6300 | 0.7915 | 0.8092 | 0.0156 | 31 |
| GPT-4.1 | Img | 71.2 | 0.39 | 0.4417 | 0.5745 | 0.5723 | 0.0142 | 29 |
| GPT-4o | Img | 68.5 | 0.39 | 0.4303 | 0.5700 | 0.5610 | 0.0134 | 23 |
| *Open-source MLLMs* | | | | | | | | |
| Qwen3-VL-235B-A22B-Thinking | Img | 75.1 | 0.38 | 0.4606 | 0.6021 | 0.5759 | - | 105 |
| Qwen3-VL-235B-A22B-Instruct | Img | 71.6 | 0.43 | 0.4279 | 0.5703 | 0.6182 | - | 35 |
| InternVL3.5-241B-A28B (thinking) | Img | 63.3 | 0.41 | 0.2856 | 0.3538 | 0.3866 | – | 29 |
| Intern-S1 (241B, thinking) | Img | 57.0 | 0.51 | 0.0664* | 0.0833* | 0.2088* | – | 39 |
| Intern-S1-mini (8B, thinking) | Img | 56.6 | 0.53 | 0.0027* | 0.0023* | 0.0158* | – | 18 |

Table 4: **Ablation on the evaluator model.** Data and correlations are identical to Tab. 3. *: The $p$-value is larger than 0.05, indicating the metric using this evaluator has no statistical significance.

slightly better than using solely semantic correctness without visual plausibility, while CLIP score and VQA score fail to capture the correctness of multidisciplinary images. In Appendix Tab. 4, we further examine the selection of MLLMs as the evaluator model.

## 4.4 ABLATION ON EVALUATOR MODELS

We further study the selection of evaluator model by examining different MLLMs, including closed-source models (GPT-5, GPT-5-mini, o3, o4-mini, GPT-4.1, GPT-4o (OpenAI, 2025a;c)) and open-source models (Qwen3-VL (Bai et al., 2025b), InternVL3.5 (Wang et al., 2025), Intern-S1, Intern-S1-mini (Bai et al., 2025a)). On the same manually evaluated samples as in Tab. 3, we run each of the models as evaluator and calculate the accuracy of answering scoring points, MAE of visual plausibility, and three correlation metrics (Kendall's $\tau$, Spearman's $\rho$, and Pearson's $r$).

As shown in Tab. 4, the latest model GPT-5 with low reasoning effort achieves the best performance with acceptable cost and time, while using medium or high reasoning effort leads to a potential decrease in performance and higher costs. Early closed-source models have relatively low performance, especially those without reasoning abilities.

For open-source models, Qwen3-VL demonstrates relative strong performance, but still lags behind GPT-5. Other models have a larger gap between closed-source models, where some scoring point accuracy is close to random guess (50%) and some correlations are even of no statistical significance, suggesting these evaluators are not applicable. This is because multidisciplinary images are still difficult for current multimodal understanding models (Yue et al., 2024b). Therefore, it is necessary to use advanced MLLMs to evaluate the generated images for higher precision and robustness.

We observe that using ground truth images as reference helps to improve the evaluation precision. We also test using captions of the ground truth images generated by GPT-5 as the ground truth, and observe that text-only ground truth is slightly worse than image-only ground truth. This is because ground truth images have better spatial/detail fidelity than textual descriptions (which often omit critical spatial details) and lower ambiguity. Using both images and textual descriptions shows no significant improvement than image-only, as images already contain sufficient detailed information and incorporating text becomes unnecessary.

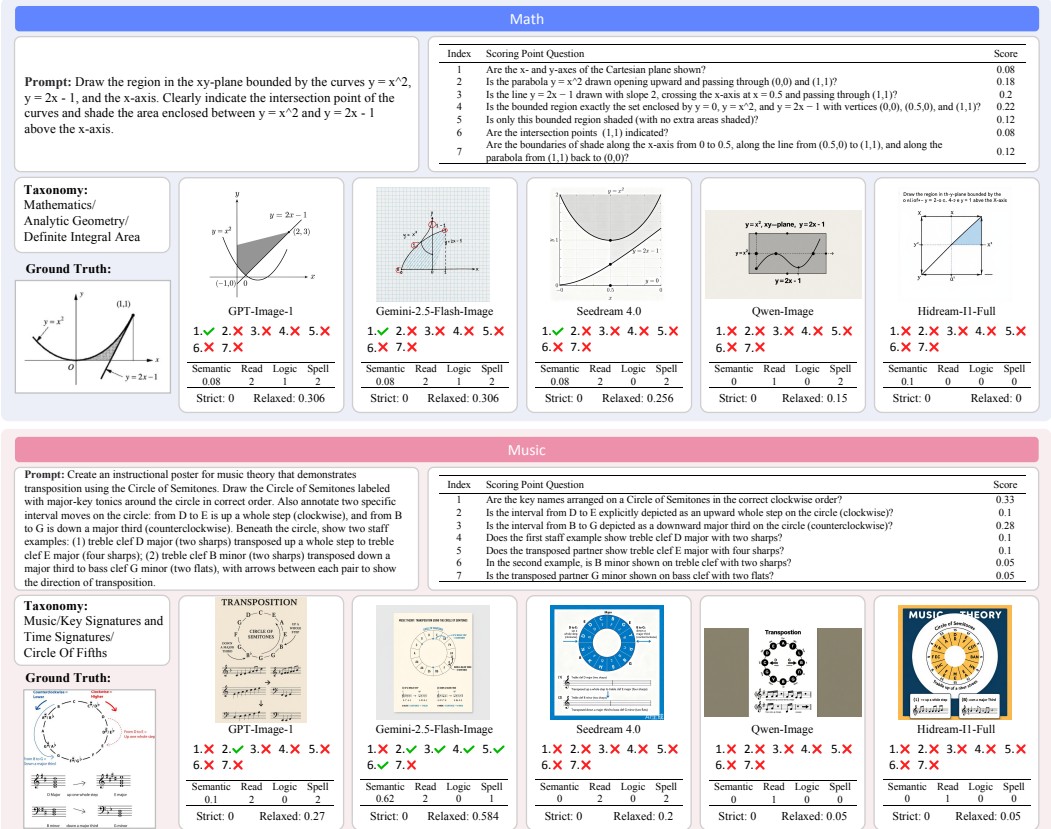

Figure 7: **Examples of images generated by top-performing models.** Semantic, Read, Logic, and Spell denote semantic correctness, logical consistency and spelling, respectively.

## 4.5 QUALITATIVE ANALYSIS

Examples of generated images are shown in Fig. 7. Taking GPT-Image-1 as an example, common error types include lack of knowledge (*e.g.* keys in the Circle of Semitones), incorrect reasoning (*e.g.* failing to infer the coordinate of the intersection point), and graph-drawing errors of logical inconsistency (*e.g.* the point $(-1, 0)$ appears to be on the y-axis), highlighting the necessity to integrate multimodal understanding, reasoning and generation. Some models show other graph-drawing errors of spelling (*e.g.* "Transpostion" for Qwen-Image) or readability (*e.g.* occluded "Music Theory" for HiDream-I1-Full), which is a key capability in multidisciplinary text-to-image exams. These examples align with our quantitative results: while top closed-source models can partially integrate basic knowledge into generation, they still struggle with precise, domain-aware execution; open-source models, on the contrary, often fail at basic aspects like generating visual plausible images. This underscores the need for future models to prioritize multidisciplinary knowledge integration, rigorous reasoning, and fine-grained visual coherence. Such failures also demonstrate the value of GenExam's fine-grained scoring criteria, as they capture flaws that broad metrics like photorealism or general aesthetics would miss. More qualitative results are provided in Appendix G.

## 5 CONCLUSION

We presented GenExam, the first benchmark for multidisciplinary text-to-image exams, designed to test whether models can integrate understanding, reasoning and generation to truly solve graph-drawing problems. GenExam offers diverse and challenging prompts with ground truth images and fine-grained scoring points, reflecting the rigor of real exams. Our experiments demonstrate that even the strongest models struggle to achieve substantial performance, highlighting the difficulty of the benchmark. We hope GenExam will guide the advancement of generative models toward expert-level intelligence in a multidisciplinary direction.

ETHICS STATEMENT

All images in our benchmark are constructed from publicly available resources and existing datasets. We will release the benchmark with clear usage guidelines to prevent misuse for generating misleading content. We have checked for potential biases across all the subjects, aiming for balanced knowledge coverage without favoring specific demographics or disciplines.

REPRODUCIBILITY STATEMENT

We ensure reproducibility by detailing GenExam's curation and evaluation protocols in Sections 3 and the Appendix, and will release the full benchmark and evaluation code publicly.

THE USE OF LARGE LANGUAGE MODELS

LLMs were only used in data curation and evaluation. No LLM was involved in the manuscript's writing.

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

# A MORE STATISTICS

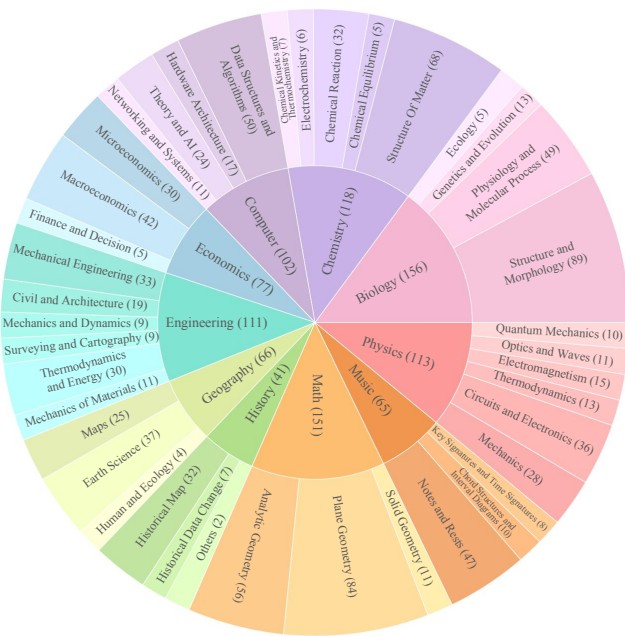

Figure 8: **Level-2 taxonomy of GenExam.** Detailed four-level taxonomy is in Sec D.

Fig. 8 shows all the level-2 taxonomy in GenExam, and Fig. 9(a) and (b) show the distribution of subjects in GenExam-Full and GenExam-Mini. We include more samples for science and engineering subjects while also covering subjects related to humanity and social science, such as Economics, Music, and History. GenExam-Mini is constructed using stratified sampling on level-3 taxonomy, which ensures similar distributions between GenExam-Full and GenExam-Mini.

The proportion of image sources is presented in Fig. 9(c), demonstrating the diversity of images in our benchmark. Word clouds of prompts in each subject are shown in Fig. 10, where we observe keywords with dense subject knowledge similar to human exams.

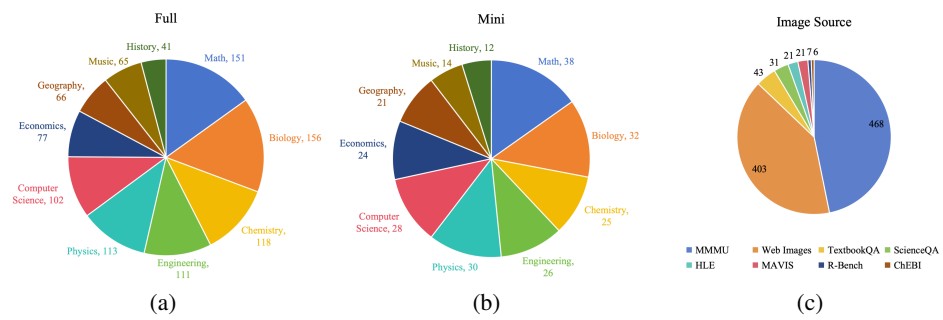

Figure 9: **More statistics:** (a) Subject distribution on GenExam-Full; (b) Subject distribution on GenExam-Mini; (c) Proportion of each image source.

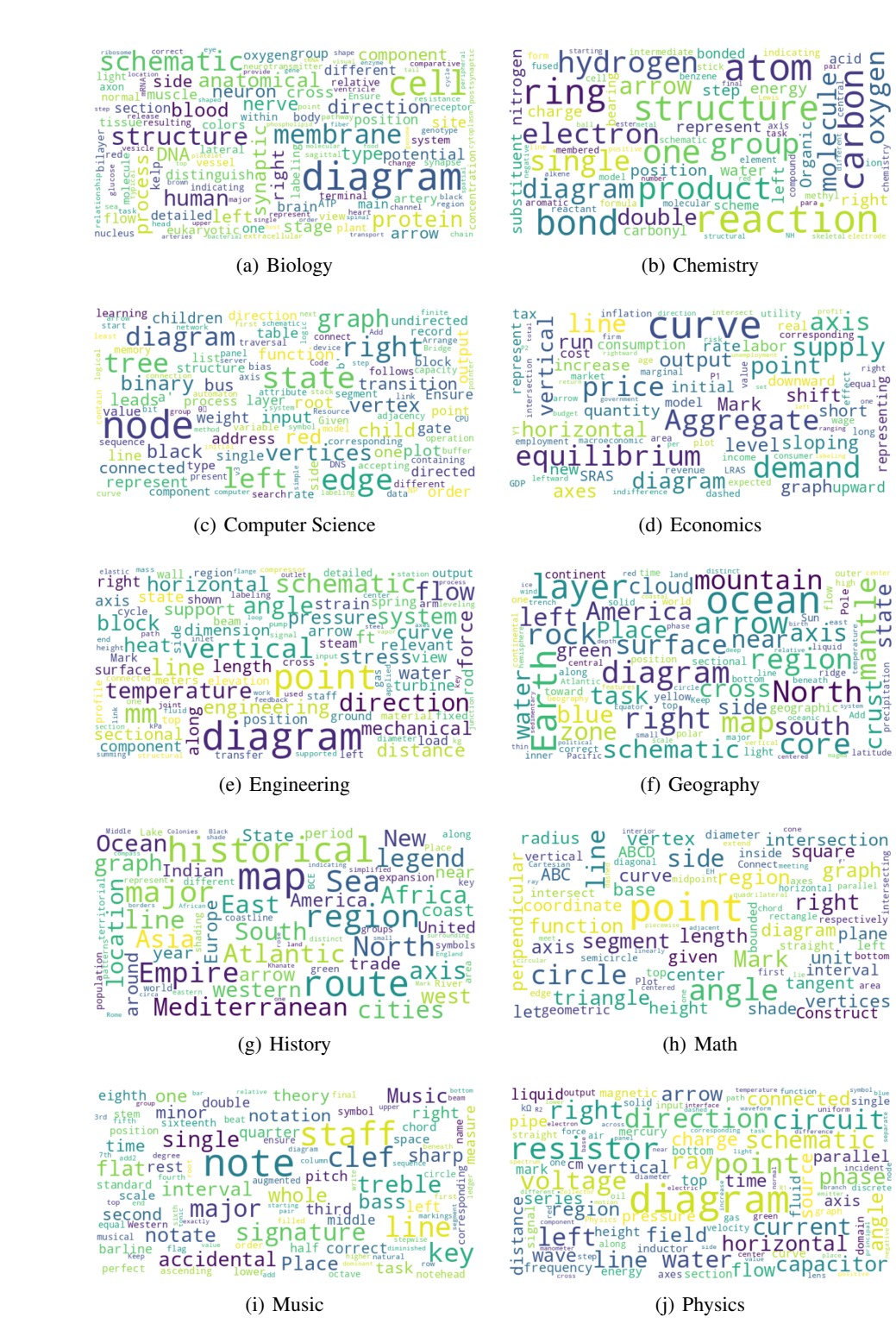

(a) Biology

(b) Chemistry

(c) Computer Science

(d) Economics

(e) Engineering

(f) Geography

(g) History

(h) Math

(i) Music

(j) Physics

Figure 10: **Word clouds of prompts in each subject.**

## B  AUTOMATIC DATA FILTERING

To obtain a high-quality dataset, we employ an automatic filtering process on the collected images. We first design heuristic rules to remove images with duplicated content, low resolution, watermarks or non-English text. Then, we leverage GPT-5 to rate each image in terms of its text richness, image domain, image complexity, and subject knowledge density. We set a threshold for each of these aspects to filter out images that do not meet the requirements of our task, including those with:

- High text richness: *e.g.* tables or screenshots of pure text. Such images are dominated by textual information and primarily examines the text rendering ability, which deviates from the core requirement of more general "image generation" for visual content.
- Undesirable image domain: *e.g.* photographs, pathological images or satellite images. These domains rely more on realistic details or professional equipment for collection, which is inconsistent with real graph-drawing questions in exams. They hardly reflect the ability to visually transform interdisciplinary knowledge.
- High complexity: *e.g.* too many components or sub-figures. Excessive components or sub-figures lead to redundant visual information, which interferes with the model's generation of core visual elements. Additionally, complex structures often exceed the model's current generation capabilities and pose overly high challenges to the model.
- Low subject knowledge density: e.g simple plots and charts without subject knowledge. Such images lack core information in interdisciplinary scenarios.

## C  ADDITIONAL EXPERIMENTS

In Tab. 5, we show the results on GenExam-Mini. The performance of the models is consistent with the results from the full dataset, which can be considered as a valid subset for efficient and affordable validation.

We provide results categorized by image types in Tab. 6. We observe that models tend to have lower performance on samples with certain image types like chemical structures, geometric shapes, sheet music and trees & graphs. These suggest potential directions for future improvements.

Results categorized by difficulty levels are presented in Tab. 7. We observe that models tend to have lower performance on samples with higher subject knowledge difficulty, demonstrating the challenge of integrate profound multidisciplinary knowledge into generation.

The results on each level-2 taxonomy is provided in Tab. 9. To highlight the difference between models, and since some models can only obtain a 0% strict score, we only report relaxed scores.

Tab. 10 shows the scores of each dimension (semantic correctness, spelling, logical consistency, readability) on each subject.

In Tab. 8, we extend the human alignment experiment to 400 additional samples covering four other models. The results show that relaxed scores maintain strong correlations with human ratings across all models, confirming the metric's generalizability. The standard deviation (std) values demonstrate high inter-rater reliability.

| Model | Strict | Relaxed |
|---|---|---|
| *Closed-source Models* | | |
| GPT-Image-1 | 12.0 | 62.4 |
| Seedream 4.0 | 8.6 | 55.1 |
| Imagen-4-Ultra | 6.5 | 53.8 |
| Gemini-2.5-Flash-Image | 4.3 | 58.7 |
| Seedream 3.0 | 0.4 | 23.4 |
| FLUX.1 Kontext max | 0.0 | 31.6 |
| *Open-source T2I Models* | | |
| Qwen-Image | 0.0 | 27.5 |
| HiDream-I1-Full | 0.0 | 22.4 |
| FLUX.1 dev | 0.0 | 16.9 |
| FLUX.1 Krea | 0.0 | 15.4 |
| Stable Diffusion 3.5 Large | 0.0 | 16.8 |
| *Open-source Unified MLLMs* | | |
| BAGEL (thinking) | 0.0 | 14.0 |
| BAGEL | 0.0 | 12.2 |
| BLIP3o-NEXT-GRPO-Text-3B | 0.0 | 12.4 |
| BLIP3o-8B | 0.0 | 7.1 |
| Show-o2-7B | 0.0 | 12.1 |
| Show-o2-1.5B-HQ | 0.0 | 11.5 |
| Janus-Pro | 0.0 | 10.1 |
| Emu3 | 0.0 | 9.0 |

Table 5: **Strict and relaxed scores on GenExam-Mini.**

| Model | Chemical Structures | | Diagrams | | Geometric Shapes | | Maps | | Plots & Charts | | Sheet Music | | Trees & Graphs | | Other | |
|---|---|---|---|---|---|---|---|---|---|---|---|---|---|---|---|---|
| | Str | Rel | Str | Rel | Str | Rel | Str | Rel | Str | Rel | Str | Rel | Str | Rel | Str | Rel |
| *Closed-source Models* | | | | | | | | | | | | | | | | |
| GPT-Image-1 | 7.1 | 45.5 | 15.6 | 66.3 | 7.3 | 54.3 | 8.0 | 68.5 | 14.1 | 64.3 | 10.0 | 54.0 | 8.3 | 52.0 | 13.8 | 61.1 |
| Seedream 4.0 | 4.8 | 38.6 | 10.8 | 59.3 | 1.6 | 40.3 | 5.4 | 54.1 | 6.5 | 53.0 | 0.0 | 35.2 | 2.8 | 48.0 | 6.9 | 54.2 |
| Gemini-2.5-Flash-Image | 4.8 | 37.1 | 5.6 | 63.4 | 1.6 | 46.3 | 3.6 | 61.2 | 2.2 | 53.1 | 1.7 | 37.5 | 2.8 | 42.9 | 3.4 | 59.6 |
| Imagen-4-Ultra | 6.0 | 35.6 | 10.5 | 60.4 | 3.2 | 37.9 | 5.4 | 57.7 | 7.2 | 52.5 | 0.0 | 39.5 | 0.0 | 32.2 | 6.9 | 47.9 |
| Seedream 3.0 | 1.2 | 14.7 | 0.0 | 26.0 | 1.6 | 23.5 | 0.0 | 34.7 | 0.0 | 14.1 | 0.0 | 22.1 | 0.0 | 17.7 | 0.0 | 20.1 |
| FLUX.1 Kontext max | 0.0 | 17.6 | 0.0 | 30.9 | 0.0 | 24.6 | 0.0 | 38.9 | 0.0 | 21.5 | 0.0 | 26.8 | 0.0 | 21.4 | 0.0 | 28.6 |
| *Open-source T2I Models* | | | | | | | | | | | | | | | | |
| Qwen-Image | 0.0 | 12.9 | 0.4 | 29.4 | 0.0 | 22.6 | 0.0 | 41.5 | 0.0 | 20.0 | 0.0 | 23.9 | 0.0 | 18.9 | 0.0 | 22.6 |
| HiDream-I1-Full | 0.0 | 10.3 | 0.0 | 22.7 | 0.0 | 21.2 | 0.0 | 34.2 | 0.0 | 17.7 | 0.0 | 21.9 | 0.0 | 17.0 | 0.0 | 20.0 |
| FLUX.1 dev | 0.0 | 13.0 | 0.0 | 17.4 | 0.0 | 11.7 | 0.0 | 30.1 | 0.0 | 11.3 | 0.0 | 22.1 | 0.0 | 14.7 | 0.0 | 12.0 |
| FLUX.1 Krea | 0.0 | 7.0 | 0.0 | 18.7 | 0.0 | 7.0 | 0.0 | 27.0 | 0.0 | 8.2 | 0.0 | 17.4 | 0.0 | 10.2 | 0.0 | 17.3 |
| Stable Diffusion 3.5 Large | 0.0 | 8.1 | 0.0 | 17.7 | 0.0 | 13.1 | 0.0 | 29.3 | 0.0 | 9.1 | 0.0 | 24.9 | 0.0 | 7.8 | 0.0 | 11.2 |
| *Open-source Unified MLLMs* | | | | | | | | | | | | | | | | |
| BAGEL (thinking) | 0.0 | 10.5 | 0.0 | 13.8 | 0.0 | 13.6 | 0.0 | 22.5 | 0.0 | 7.5 | 0.0 | 15.5 | 0.0 | 8.4 | 0.0 | 9.0 |
| BAGEL | 0.0 | 7.9 | 0.0 | 11.7 | 0.0 | 17.1 | 0.0 | 19.1 | 0.0 | 6.4 | 0.0 | 14.8 | 0.0 | 9.8 | 0.0 | 6.8 |
| Show-o2-7B | 0.0 | 4.3 | 0.0 | 13.1 | 0.0 | 10.6 | 0.0 | 22.8 | 0.0 | 7.2 | 0.0 | 8.9 | 0.0 | 3.8 | 0.0 | 9.3 |
| Show-o2-1.5B-HQ | 0.0 | 4.4 | 0.0 | 11.5 | 0.0 | 6.8 | 0.0 | 22.6 | 0.0 | 7.3 | 0.0 | 7.8 | 0.0 | 4.3 | 0.0 | 7.8 |
| BLIP3o-NEXT-GRPO-Text-3B | 0.0 | 7.8 | 0.0 | 12.9 | 0.0 | 16.6 | 0.0 | 16.7 | 0.0 | 8.6 | 0.0 | 16.4 | 0.0 | 13.4 | 0.0 | 14.9 |
| BLIP3o-8B | 0.0 | 4.3 | 0.0 | 7.2 | 0.0 | 8.1 | 0.0 | 16.5 | 0.0 | 2.6 | 0.0 | 6.5 | 0.0 | 3.4 | 0.0 | 5.6 |
| Janus-Pro | 0.0 | 9.3 | 0.0 | 9.5 | 0.0 | 15.4 | 0.0 | 11.9 | 0.0 | 4.3 | 0.0 | 15.0 | 0.0 | 4.7 | 0.0 | 11.4 |
| Emu3 | 0.0 | 0.0 | 0.0 | 12.7 | 0.0 | 5.0 | 0.0 | 13.7 | 0.0 | 4.0 | 0.0 | 5.8 | 0.0 | 30.0 | 0.0 | 0.0 |

Table 6: **Strict scores (Str) and relaxed scores (Rel) on GenExam for different image types.**

| Model | Easy | | Medium | | Hard | |
|---|---|---|---|---|---|---|
| | Strict | Relaxed | Strict | Relaxed | Strict | Relaxed |
| *Closed-source Models* | | | | | | |
| GPT-Image-1 | 18.6 | 68.1 | 12.8 | 62.2 | 9.9 | 58.5 |
| Seedream 4.0 | 11.2 | 57.8 | 5.5 | 50.3 | 7.9 | 53.2 |
| Gemini-2.5-Flash-Image | 5.4 | 58.4 | 3.2 | 55.5 | 4.7 | 55.4 |
| Imagen-4-Ultra | 9.9 | 58.6 | 7.1 | 50.7 | 7.1 | 51.8 |
| Seedream 3.0 | 0.4 | 31.3 | 0.3 | 21.8 | 0.0 | 19.0 |
| FLUX.1 Kontext max | 0.0 | 37.3 | 0.0 | 26.1 | 0.0 | 23.8 |
| *Open-source T2I Models* | | | | | | |
| Qwen-Image | 0.4 | 34.6 | 0.3 | 25.4 | 0.0 | 21.3 |
| HiDream-I1-Full | 0.0 | 29.7 | 0.0 | 19.8 | 0.0 | 17.1 |
| FLUX.1 dev | 0.0 | 25.2 | 0.0 | 15.4 | 0.0 | 12.3 |
| FLUX.1 Krea | 0.0 | 24.9 | 0.0 | 13.0 | 0.0 | 12.1 |
| Stable Diffusion 3.5 Large | 0.0 | 25.0 | 0.0 | 14.9 | 0.0 | 11.2 |
| *Open-source Unified MLLMs* | | | | | | |
| BAGEL (thinking) | 0.0 | 21.1 | 0.0 | 12.8 | 0.0 | 7.7 |
| BAGEL | 0.0 | 17.9 | 0.0 | 11.5 | 0.0 | 7.1 |
| Show-o2-7B | 0.0 | 18.1 | 0.0 | 10.2 | 0.0 | 7.9 |
| Show-o2-1.5B-HQ | 0.0 | 17.1 | 0.0 | 8.7 | 0.0 | 6.9 |
| BLIP3o-NEXT-GRPO-Text-3B | 0.0 | 20.2 | 0.0 | 12.3 | 0.0 | 8.1 |
| BLIP3o-8B | 0.0 | 10.6 | 0.0 | 5.8 | 0.0 | 5.1 |
| Janus-Pro | 0.0 | 13.6 | 0.0 | 9.9 | 0.0 | 6.5 |
| Emu3 | 0.0 | 14.2 | 0.0 | 7.5 | 0.0 | 6.0 |

Table 7: **Results categorized by subject knowledge difficulty.**

| Model | Semantic | | Spelling | | Logic | | Readability | | Kendall $\tau$ | Spearman $\rho$ | Pearson $r$ |
|---|---|---|---|---|---|---|---|---|---|---|---|
| | MAE | std | MAE | std | MAE | std | MAE | std | | | |
| GPT-Image-1 | 0.104 | 0.0197 | 0.112 | 0.0563 | 0.276 | 0.0851 | 0.201 | 0.0642 | 0.6746 | 0.8217 | 0.8444 |
| Seedream 4.0 | 0.113 | 0.0234 | 0.117 | 0.0678 | 0.287 | 0.1025 | 0.219 | 0.0793 | 0.6628 | 0.8061 | 0.8054 |
| Gemini-2.5-Flash-Image | 0.119 | 0.0248 | 0.108 | 0.0623 | 0.283 | 0.0964 | 0.211 | 0.0751 | 0.6537 | 0.7969 | 0.8062 |
| Qwen-Image | 0.132 | 0.0389 | 0.123 | 0.1152 | 0.308 | 0.1437 | 0.229 | 0.1186 | 0.6274 | 0.7627 | 0.7685 |
| FLUX.1 dev | 0.156 | 0.0427 | 0.159 | 0.1284 | 0.328 | 0.1589 | 0.246 | 0.1325 | 0.6079 | 0.7653 | 0.7564 |

Table 8: **Human alignment on more models.**

| Level-2 Taxonomy | GPT-Image-1 | Seedream 4.0 | Gemini-2.5-Flash-Image | Imagen-4-Ultra | FLUX.1 Kontext max | Qwen-Image | HiDream-I1-Full | FLUX.1 dev | Stable Diffusion 3.5 Large | BAGEL (thinking) | Show-o2-7B | BLIP3o-NEXT-GRPO-Text-3B |
|---|---|---|---|---|---|---|---|---|---|---|---|---|
| Biology/Ecology | 47.1 | 36.3 | 53.0 | 77.5 | 17.1 | 19.7 | 24.0 | 21.7 | 11.4 | 21.4 | 8.7 | 7.2 |
| Biology/Genetics and Evolution | 66.8 | 69.1 | 72.8 | 57.7 | 24.9 | 22.3 | 10.6 | 9.8 | 11.5 | 8.4 | 6.2 | 7.5 |
| Biology/Physiology and Molecular Process | 73.0 | 69.1 | 71.9 | 62.1 | 28.7 | 23.2 | 20.5 | 16.0 | 15.2 | 9.3 | 8.6 | 7.3 |
| Biology/Structure and Morphology | 76.5 | 74.2 | 74.1 | 72.4 | 46.8 | 39.0 | 33.6 | 28.6 | 27.5 | 19.1 | 16.2 | 21.7 |
| Chemistry/Chemical Equilibrium | 88.7 | 80.6 | 60.5 | 50.8 | 26.8 | 29.1 | 15.1 | 20.4 | 36.4 | 15.4 | 12.4 | 15.4 |
| Chemistry/Chemical Kinetics and Thermochemistry | 87.8 | 77.3 | 71.5 | 73.5 | 20.6 | 34.9 | 22.1 | 15.7 | 8.4 | 11.6 | 1.4 | 6.4 |
| Chemistry/Chemical Reaction | 34.2 | 33.7 | 26.4 | 30.2 | 10.4 | 3.5 | 4.9 | 3.9 | 5.0 | 2.3 | 0.3 | 2.0 |
| Chemistry/Electrochemistry | 74.4 | 78.3 | 79.8 | 70.4 | 20.1 | 24.1 | 16.8 | 10.1 | 15.3 | 12.3 | 2.2 | 11.3 |
| Chemistry/Structure Of Matter | 54.5 | 43.5 | 47.4 | 45.4 | 22.6 | 17.0 | 16.2 | 15.8 | 11.3 | 16.2 | 6.9 | 12.2 |
| Computer/Data Structures and Algorithms | 52.1 | 48.0 | 42.2 | 34.0 | 20.9 | 20.3 | 17.4 | 12.6 | 6.9 | 7.6 | 3.9 | 11.6 |
| Computer/Hardware Architecture | 48.2 | 49.4 | 48.2 | 41.7 | 14.8 | 20.1 | 12.2 | 5.8 | 7.9 | 2.4 | 3.0 | 2.4 |
| Computer/Networking and Systems | 59.8 | 57.0 | 50.4 | 50.6 | 19.1 | 10.9 | 12.9 | 9.5 | 3.1 | 1.9 | 2.7 | 0.9 |
| Computer/Theory and AI | 66.2 | 60.5 | 56.1 | 46.6 | 25.9 | 19.1 | 14.5 | 11.8 | 6.9 | 7.9 | 8.7 | 8.7 |
| Economics/Finance and Decision | 74.4 | 65.0 | 62.6 | 47.9 | 26.9 | 32.4 | 17.2 | 18.8 | 8.6 | 13.5 | 7.0 | 9.0 |
| Economics/Macroeconomics | 65.5 | 57.8 | 60.0 | 59.4 | 24.9 | 19.9 | 21.0 | 8.2 | 6.4 | 5.2 | 7.7 | 6.0 |
| Economics/Microeconomics | 64.8 | 52.0 | 59.0 | 62.1 | 18.0 | 18.9 | 15.9 | 8.8 | 10.1 | 6.7 | 6.1 | 11.0 |
| Engineering/Civil and Architecture | 66.4 | 62.4 | 60.5 | 64.7 | 33.3 | 34.9 | 30.2 | 18.2 | 21.8 | 13.9 | 16.0 | 13.8 |
| Engineering/Mechanical Engineering | 64.4 | 54.3 | 62.5 | 68.3 | 31.8 | 38.0 | 27.4 | 17.0 | 16.3 | 15.5 | 11.5 | 11.1 |
| Engineering/Mechanics and Dynamics | 61.7 | 51.1 | 48.3 | 45.6 | 11.1 | 15.3 | 9.7 | 0.0 | 2.3 | 0.6 | 8.5 | 2.6 |
| Engineering/Mechanics of Materials | 69.8 | 62.2 | 71.0 | 49.9 | 20.4 | 17.2 | 21.8 | 15.3 | 18.3 | 8.7 | 4.7 | 6.0 |
| Engineering/Surveying and Cartography | 74.5 | 65.4 | 79.5 | 72.0 | 37.1 | 37.8 | 33.7 | 11.6 | 16.9 | 12.8 | 20.9 | 15.3 |
| Engineering/Thermodynamics and Energy | 63.1 | 64.8 | 72.0 | 72.9 | 29.1 | 32.4 | 20.0 | 12.7 | 16.0 | 6.6 | 10.3 | 8.9 |
| Geography/Earth Science | 77.9 | 72.9 | 73.5 | 75.7 | 53.1 | 56.0 | 38.5 | 37.9 | 41.1 | 30.2 | 37.6 | 25.3 |
| Geography/Human and Ecology | 61.9 | 47.2 | 61.8 | 36.1 | 18.0 | 20.4 | 17.3 | 8.5 | 6.4 | 1.2 | 6.2 | 4.2 |
| Geography/Maps | 69.8 | 56.3 | 66.7 | 58.9 | 44.0 | 44.7 | 35.7 | 38.8 | 40.5 | 30.2 | 31.2 | 24.5 |
| History/Historical Data Change | 62.3 | 75.2 | 53.1 | 57.9 | 25.3 | 31.0 | 25.3 | 10.5 | 6.9 | 15.5 | 8.6 | 12.2 |
| History/Historical Map | 68.6 | 51.3 | 57.5 | 56.4 | 36.3 | 40.5 | 33.9 | 24.3 | 21.3 | 16.3 | 15.9 | 9.9 |
| History/Others | 65.0 | 79.1 | 64.8 | 78.8 | 18.3 | 34.8 | 19.5 | 19.8 | 5.3 | 12.8 | 13.1 | 7.8 |
| Mathematics/Analytic Geometry | 49.1 | 36.6 | 37.8 | 28.7 | 20.7 | 16.4 | 12.4 | 10.7 | 11.3 | 6.5 | 10.0 | 14.5 |
| Mathematics/Plane Geometry | 52.1 | 40.2 | 45.6 | 36.2 | 24.1 | 18.2 | 18.3 | 12.8 | 11.8 | 14.4 | 11.0 | 15.3 |
| Mathematics/Solid Geometry | 65.1 | 52.6 | 50.6 | 70.5 | 32.7 | 37.3 | 27.3 | 16.1 | 19.3 | 16.7 | 13.5 | 22.0 |
| Music/Chord Structures and Interval Diagrams | 36.6 | 31.2 | 32.6 | 21.2 | 12.2 | 8.1 | 13.5 | 9.1 | 6.3 | 1.0 | 1.0 | 5.9 |
| Music/Key Signatures and Time Signatures | 43.0 | 36.2 | 41.0 | 28.6 | 27.3 | 20.4 | 20.7 | 28.2 | 30.4 | 12.9 | 7.1 | 16.2 |
| Music/Notes and Rests | 57.7 | 34.9 | 37.2 | 43.7 | 27.9 | 27.2 | 23.1 | 22.8 | 26.8 | 17.9 | 10.7 | 16.9 |
| Physics/Circuits and Electronics | 55.1 | 36.8 | 47.3 | 39.1 | 14.5 | 14.7 | 8.8 | 9.7 | 5.0 | 7.2 | 4.6 | 3.7 |
| Physics/Electromagnetism | 71.9 | 60.0 | 66.0 | 66.0 | 29.3 | 35.1 | 26.0 | 19.1 | 19.1 | 24.6 | 18.2 | 20.9 |
| Physics/Mechanics | 74.9 | 47.4 | 79.3 | 71.5 | 33.1 | 37.7 | 23.6 | 11.4 | 16.8 | 14.0 | 16.2 | 15.6 |
| Physics/Optics and Waves | 74.5 | 66.2 | 63.0 | 69.8 | 38.3 | 28.8 | 24.8 | 36.0 | 21.5 | 20.7 | 19.0 | 7.9 |
| Physics/Quantum Mechanics | 63.4 | 60.1 | 58.8 | 64.8 | 35.7 | 28.6 | 25.1 | 20.7 | 26.9 | 24.1 | 19.5 | 13.3 |
| Physics/Thermodynamics | 68.8 | 50.6 | 53.2 | 51.6 | 17.7 | 22.9 | 8.2 | 14.0 | 3.4 | 5.1 | 7.5 | 7.8 |

Table 9: **Relaxed scores on level-2 Taxonomy.**

| Subject | Dimension | GPT-Image-1 | Seedream 4.0 | Gemini-2.5-Flash-Image | Imagen-4-Ultra | FLUX.1 Kontext max | Qwen-Image | HiDream-I1-Full | FLUX.1 dev | Stable Diffusion 3.5 Large | BAGEL (thinking) | Show-o2-7B | BLIP3o-NEXT-GRPO-Text-3B |
|---|---|---|---|---|---|---|---|---|---|---|---|---|---|
| Biology | Semantic Correctness | 0.72 | 0.70 | 0.76 | 0.69 | 0.40 | 0.32 | 0.26 | 0.20 | 0.22 | 0.10 | 0.12 | 0.14 |
| | Spelling | 1.53 | 1.50 | 0.93 | 1.21 | 0.49 | 0.28 | 0.25 | 0.29 | 0.22 | 0.17 | 0.04 | 0.13 |
| | Logical Consistency | 1.32 | 1.13 | 1.24 | 1.17 | 0.50 | 0.51 | 0.31 | 0.43 | 0.40 | 0.39 | 0.40 | 0.29 |
| | Readability | 1.74 | 1.71 | 1.72 | 1.63 | 1.15 | 1.15 | 1.24 | 1.07 | 0.69 | 1.01 | 0.37 | 0.67 |
| Chemistry | Semantic Correctness | 0.46 | 0.41 | 0.43 | 0.43 | 0.15 | 0.14 | 0.10 | 0.07 | 0.08 | 0.08 | 0.04 | 0.06 |
| | Spelling | 1.66 | 1.31 | 0.94 | 1.06 | 0.51 | 0.23 | 0.27 | 0.40 | 0.35 | 0.37 | 0.03 | 0.25 |
| | Logical Consistency | 0.86 | 0.65 | 0.57 | 0.57 | 0.19 | 0.14 | 0.12 | 0.24 | 0.27 | 0.37 | 0.21 | 0.26 |
| | Readability | 1.72 | 1.50 | 1.59 | 1.27 | 1.05 | 0.77 | 0.86 | 0.82 | 0.33 | 0.58 | 0.19 | 0.53 |
| Economy | Semantic Correctness | 0.64 | 0.52 | 0.62 | 0.61 | 0.21 | 0.20 | 0.18 | 0.07 | 0.09 | 0.05 | 0.07 | 0.06 |
| | Spelling | 1.66 | 1.71 | 1.36 | 1.34 | 0.42 | 0.39 | 0.19 | 0.29 | 0.08 | 0.00 | 0.01 | 0.17 |
| | Logical Consistency | 1.06 | 0.64 | 0.65 | 0.78 | 0.19 | 0.16 | 0.19 | 0.08 | 0.06 | 0.19 | 0.29 | 0.29 |
| | Readability | 1.53 | 1.52 | 1.32 | 1.27 | 0.95 | 0.69 | 0.88 | 0.48 | 0.21 | 0.36 | 0.14 | 0.27 |
| Engineering | Semantic Correctness | 0.62 | 0.57 | 0.66 | 0.67 | 0.26 | 0.33 | 0.19 | 0.12 | 0.15 | 0.08 | 0.11 | 0.08 |
| | Spelling | 1.64 | 1.60 | 1.30 | 1.36 | 0.67 | 0.50 | 0.66 | 0.32 | 0.43 | 0.23 | 0.12 | 0.24 |
| | Logical Consistency | 1.17 | 0.89 | 1.05 | 1.01 | 0.42 | 0.41 | 0.35 | 0.21 | 0.27 | 0.34 | 0.41 | 0.18 |
| | Readability | 1.64 | 1.49 | 1.55 | 1.41 | 1.02 | 0.92 | 1.15 | 0.60 | 0.46 | 0.48 | 0.31 | 0.44 |
| Geography | Semantic Correctness | 0.72 | 0.64 | 0.71 | 0.68 | 0.48 | 0.49 | 0.30 | 0.29 | 0.34 | 0.20 | 0.32 | 0.21 |
| | Spelling | 1.51 | 1.29 | 1.08 | 1.08 | 0.62 | 0.65 | 0.73 | 0.77 | 0.80 | 0.68 | 0.39 | 0.30 |
| | Logical Consistency | 1.41 | 1.03 | 1.30 | 1.18 | 0.83 | 0.91 | 0.82 | 0.94 | 1.02 | 0.91 | 0.92 | 0.59 |
| | Readability | 1.80 | 1.71 | 1.73 | 1.55 | 1.27 | 1.45 | 1.45 | 1.53 | 1.12 | 1.24 | 0.83 | 0.97 |
| History | Semantic Correctness | 0.66 | 0.54 | 0.59 | 0.60 | 0.32 | 0.41 | 0.32 | 0.14 | 0.17 | 0.11 | 0.14 | 0.09 |
| | Spelling | 1.41 | 1.32 | 0.93 | 0.83 | 0.40 | 0.39 | 0.27 | 0.39 | 0.27 | 0.15 | 0.12 | 0.02 |
| | Logical Consistency | 1.15 | 0.85 | 0.78 | 0.80 | 0.55 | 0.39 | 0.39 | 0.71 | 0.34 | 0.66 | 0.49 | 0.15 |
| | Readability | 1.63 | 1.68 | 1.49 | 1.46 | 1.20 | 1.24 | 1.20 | 1.34 | 0.68 | 0.90 | 0.34 | 0.56 |
| Math | Semantic Correctness | 0.43 | 0.31 | 0.39 | 0.30 | 0.15 | 0.13 | 0.09 | 0.06 | 0.07 | 0.04 | 0.08 | 0.05 |
| | Spelling | 1.77 | 1.60 | 1.41 | 1.38 | 1.12 | 0.81 | 0.83 | 0.54 | 0.79 | 0.53 | 0.18 | 0.90 |
| | Logical Consistency | 0.94 | 0.54 | 0.48 | 0.40 | 0.36 | 0.19 | 0.27 | 0.26 | 0.26 | 0.58 | 0.52 | 0.62 |
| | Readability | 1.62 | 1.49 | 1.21 | 1.16 | 1.08 | 0.90 | 0.93 | 0.77 | 0.44 | 0.59 | 0.38 | 0.81 |
| Music | Semantic Correctness | 0.42 | 0.25 | 0.24 | 0.27 | 0.13 | 0.13 | 0.08 | 0.10 | 0.12 | 0.07 | 0.05 | 0.06 |
| | Spelling | 1.74 | 1.42 | 1.46 | 1.51 | 1.38 | 0.91 | 0.88 | 0.78 | 1.18 | 0.86 | 0.57 | 0.63 |
| | Logical Consistency | 0.98 | 0.45 | 0.77 | 0.74 | 0.51 | 0.42 | 0.75 | 0.71 | 1.09 | 0.23 | 0.18 | 0.52 |
| | Readability | 1.89 | 1.49 | 1.74 | 1.63 | 1.42 | 1.49 | 1.54 | 1.40 | 0.80 | 0.86 | 0.28 | 1.09 |
| Physics | Semantic Correctness | 0.62 | 0.43 | 0.61 | 0.57 | 0.22 | 0.25 | 0.12 | 0.12 | 0.11 | 0.09 | 0.12 | 0.07 |
| | Spelling | 1.74 | 1.62 | 1.28 | 1.32 | 0.69 | 0.45 | 0.52 | 0.19 | 0.45 | 0.42 | 0.16 | 0.36 |
| | Logical Consistency | 1.16 | 0.65 | 0.82 | 0.83 | 0.34 | 0.42 | 0.25 | 0.29 | 0.22 | 0.42 | 0.31 | 0.30 |
| | Readability | 1.75 | 1.45 | 1.56 | 1.36 | 1.00 | 0.91 | 1.06 | 0.70 | 0.36 | 0.66 | 0.26 | 0.52 |
| Computer Science | Semantic Correctness | 0.48 | 0.47 | 0.45 | 0.38 | 0.16 | 0.16 | 0.11 | 0.07 | 0.05 | 0.03 | 0.05 | 0.03 |
| | Spelling | 1.75 | 1.73 | 1.35 | 1.25 | 0.87 | 0.66 | 0.52 | 0.32 | 0.35 | 0.29 | 0.05 | 0.51 |
| | Logical Consistency | 1.05 | 0.71 | 0.58 | 0.47 | 0.23 | 0.21 | 0.25 | 0.27 | 0.10 | 0.24 | 0.14 | 0.28 |
| | Readability | 1.66 | 1.44 | 1.30 | 1.04 | 0.84 | 0.75 | 0.81 | 0.58 | 0.18 | 0.29 | 0.11 | 0.45 |

Table 10: **Scores of each dimension for all subjects.**

# D FULL TAXONOMY LIST

In the table below, we show the complete list of four-level taxonomy and numbers of samples (denoted in parentheses). Blue text indicates mapping with ISCED-F (UNESCO, 2013) codes.

| Level 1 | Level 2 | Level 3 | Level 4 |
|---|---|---|---|
| Biology (156) | Ecology (5) 0521 Environmental sciences | Ecosystem (5) | Food Chain and Food Web (4) |
| | | | Niche (1) |
| | Genetics and Evolution (13) 0511 Biology | Evolution and Population Genetics (3) | Evolutionary Tree (2) |
| | | | Pedigree Dominant and Recessive Genetic Diseases (1) |
| | | Genetics (10) | Gene Linkage Map (1) |
| | | | Gene Structure (1) |
| | | | Mendelian Genetics (7) |
| | | | Transcription and Splicing Mechanisms (1) |
| | Physiology and Molecular Process (49) 0511 Biology 0512 Biochemistry 0912 Medicine 0916 Pharmacy | Cell Physiology (3) | Cell Division (3) |
| | | Molecular Mechanism (4) | Metabolism (1) |
| | | | Microbiology (1) |
| | | | Signaling and Regulation (2) |
| | | Molecular Mechanisms (15) | Genetic Information Transmission (12) |
| | | | Material Metabolism (3) |
| | | Pathology and Pharmacology (6) | General Adaptation Syndrome (1) |
| | | | Infection Spread (2) |
| | | | Pharmacological Dose Response Curve (1) |
| | | | Stress Response Model (1) |
| | | | Tumor and Inflammation (1) |
| | | Systemic Physiology (21) | Blood Coagulation (1) |
| | | | Germ Layer Differentiation (1) |
| | | | Immunity (3) |
| | | | Nervous System (13) |
| | | | Photosynthesis (2) |
| | | | Respiration and Gas Exchange (1) |
| | Structure and Morphology (89) 0511 Biology 0512 Biochemistry | Cell Structure (27) | Basic Cell Structure (18) |
| | | | Microbial Morphology (4) |
| | | | Special Cells (5) |
| | | Molecular Structure (6) | Biomacromolecules (4) |
| | | | Biomolecules (2) |
| | | Organ Structure (48) | Anal Canal (1) |
| | | | Bladder (1) |
| | | | Brain (16) |
| | | | Digestive Tract (3) |
| | | | Ear (1) |
| | | | Eye (4) |
| | | | Heart (4) |
| | | | Joint Structure (3) |
| | | | Kidney (2) |
| | | | Larynx (1) |
| | | | Lung (1) |
| | | | Muscular and Skeletal System (2) |
| | | | Nervous System (1) |
| | | | Pancreas (1) |
| | | | Plant Seed (1) |
| | | | Plant Stem Tissue (2) |
| | | | Skin (1) |
| | | | Vascular System (3) |
| | | Others (2) | Optical Microscope Structure (2) |
| | | Tissue Structure (6) | Epithelial Tissue (3) |
| | | | Plant Vascular Tissue (3) |

| Level 1 | Level 2 | Level 3 | Level 4 |
|---|---|---|---|
| Chemistry (118) 0531 Chemistry | Chemical Equilibrium (5) | Acid Base Titration (2) | |
| | | Solubility (3) | |
| | Chemical Kinetics and Thermochemistry (7) | | |
| | Chemical Reaction (32) | Inorganic Reaction (2) | |
| | | Organic Reaction (27) | |
| | | Reaction Mechanism (3) | |
| | Electrochemistry (6) | | |
| | Structure Of Matter (68) | Atomic Structure (14) | Atomic Model (6) |
| | | | Electron Configuration (8) |
| | | Crystal Structure (3) | |
| | | Element Abundance Distribution (2) | |
| | | Molecular Structure (49) | Electron Configuration and Intermolecular Forces (8) |
| | | | Isomers (4) |
| | | | Molecular Structure Diagram (11) |
| | | | Organic Compound (26) |
| Computer (102) | Data Structures and Algorithms (50) 0613 Software and applications development and analysis | ER Diagram (2) | |
| | | Graph (24) | Adjacency List (3) |
| | | | Directed Graph (5) |
| | | | Graph Algorithms (4) |
| | | | Maximum Flow (1) |
| | | | Shortest Path (1) |
| | | | Undirected Graph (10) |
| | | Linked List (2) | |
| | | Queue (1) | |
| | | Sorting (2) | |
| | | Tree (18) | B Tree (1) |
| | | | Others (2) |
| | | | Search Tree (7) |
| | | | Syntax Tree (2) |
| | | | Traversal (6) |
| | Hardware Architecture (17) 0613 Software and applications development and analysis | Bus Structure (3) | |
| | | Cache (3) | Others (1) |
| | | Digital Circuits (7) | Logic Gates (5) |
| | | | Multiplexer Application (2) |
| | | Instruction Format (1) | |
| | | Pipeline (3) | |
| | Networking and Systems (11) 0612 Database and network design and administration | Computer Networks (7) | Packet Structure (3) |
| | | | TCP and IP Protocol Stack (2) |
| | | | Topology (2) |
| | | Operating System (4) | Deadlock Resource Allocation Diagram (1) |
| | | | Memory Paging (2) |
| | | | Process Scheduling (1) |
| | Theory and AI (24) 0613 Software and applications development and analysis 0619 Artificial Intelligence | Compiler Principles (2) | |
| | | Finite Automaton (8) | |
| | | Machine Learning (14) | Learning Rate Impact (2) |
| | | | Multicollinearity (1) |
| | | | Neural Networks (3) |
| | | | Overfitting and Underfitting Diagram (1) |
| | | | ROC Curve (1) |
| | | | Sampling Methods (1) |
| | | | Training and Testing Curves (4) |
| | | | Variance Bias Tradeoff (1) |

| Level 1 | Level 2 | Level 3 | Level 4 |
|---|---|---|---|
| Economics (77) | Finance and Decision (5) 0412 Finance, banking and insurance | CAPM (2) | Beta Coefficient Scatter Plot (1) |
| | | Risk Return Chart (3) | |
| | Macroeconomics (42) 0311 Economics | AD AS Model Aggregate Demand and Aggregate Supply (29) | |
| | | Business Cycle (1) | |
| | | Equilibrium Unemployment (1) | |
| | | Exchange Rates and Monetary Policy (1) | |
| | | IS LM Model (1) | |
| | | Laffer Curve (1) | |
| | | Lorenz Curve (1) | |
| | | Other Economic Statistical Charts (3) | |
| | | Other Macroeconomic Theories (1) | |
| | | Phillips Curve Inflation and Unemployment (3) | |
| | Microeconomics (30) 0311 Economics | Cost Curves (4) | |
| | | Expected Utility Curve (1) | |
| | | Game Theory (1) | |
| | | PPF and Utility Curves (7) | Budget Constraint Line (1) |
| | | | Indifference Curve Utility Maximization (6) |
| | | Supply Demand Curves (17) | Consumer and Producer Surplus (2) |
| | | | Elasticity Price and Income (4) |
| | | | Equilibrium Price (9) |
| | | | Price Ceiling (1) |
| | | | Tax Impact (1) |
| Engineering (111) | Civil and Architecture (19) 0731 Architecture and town planning 0732 Building and civil engineering | Building Structure (4) | |
| | | Engineering Drawings (11) | Detail Drawings (4) |
| | | | Plan View (1) |
| | | | Sectional and Profile Views (6) |
| | | Geotechnical Engineering (3) | |
| | | Roads and Bridges (1) | |
| | Mechanical Engineering (33) 0715 Mechanics and metal trades | Engineering Drawings (32) | Schematic Diagram (18) |
| | | | Section View (14) |
| | | Machine Parts (1) | Simple Parts (1) |
| | Mechanics and Dynamics (9) 0715 Mechanics and metal trades | Vibration and Control (9) | Block Diagram (4) |
| | | | Damping and Vibration (5) |
| | Mechanics of Materials (11) 0715 Mechanics and metal trades | Stress Strain Relationship (11) | |
| | Surveying and Cartography (9) 0715 Mechanics and metal trades | Data Processing and Adjustment (3) | |
| | | Maps and Diagrams (2) | |
| | | Sectional and Profile Views (4) | |
| | Thermodynamics and Energy (30) 0713 Electricity and energy | Energy (2) | |
| | | Energy Changes (5) | |
| | | States and Phase Changes (10) | |
| | | Thermal Cycle (13) | |
| Geography (66) | Earth Science (37) 0532 Earth sciences | Astronomy (3) | |
| | | Climate and Circulation (16) | Atmospheric Circulation (2) |
| | | | Climate Chart (2) |
| | | | Energy Balance (1) |
| | | | Temperature Change (1) |
| | | | Water Cycle (10) |
| | | Geological Age (1) | |
| | | Geomagnetic Field (1) | |
| | | Greenhouse Effect (1) | |
| | | Landforms and Geology (14) | |
| | | Latitude and Longitude (1) | |

| Level 1 | Level 2 | Level 3 | Level 4 |
|---|---|---|---|
| | Human and Ecology (4) 0314 Sociology and cultural studies | Population and City (3) | |
| | | Urban Planning Map (1) | |
| | Maps (25) 0314 Sociology and cultural studies | Earthquake Belt Distribution (2) | |
| | | Tropical Temperate and Frigid Zones (2) | |
| | | World and Regional Map (21) | |
| History (41) 0222 History and archaeology | Historical Data Change (7) | Others (2) | |
| | | Population (3) | |
| | | Unemployment Rate (2) | |
| | Historical Map (32) | Route Map (12) | Others (6) |
| | | | Trade (6) |
| | | Territory Map (20) | |
| | Others (2) | | |
| Mathematics (151) 0541 Mathematics | Analytic Geometry (56) | Absolute Value Function (1) | |
| | | Definite Integral Area (12) | |
| | | Exponential and Logarithmic Function (2) | |
| | | Geometric Meaning Of Derivative (1) | |
| | | Inequality Region (2) | Linear Programming (2) |
| | | Linear Function (3) | |
| | | Other Function (14) | |
| | | Parametric Equation and Polar Curve (4) | |
| | | Piecewise Function (8) | |
| | | Quadratic Function (6) | |
| | | Trigonometric Function (3) | |
| | Plane Geometry (84) | Angle (3) | |
| | | Circle (22) | Chord (8) |
| | | | Inscribed and Circumscribed Circle (6) |
| | | | Others (4) |
| | | | Tangent (4) |
| | | Complex Geometry Problem (23) | |
| | | Rectangle and Polygon (25) | Other Polygon (2) |
| | | | Other Quadrilateral (5) |
| | | | Parallelogram (2) |
| | | | Pentagon (2) |
| | | | Rectangle (10) |
| | | | Regular Hexagon (1) |
| | | | Trapezoid (3) |
| | | Triangle (11) | Altitude (1) |
| | | | Angle Bisector (1) |
| | | | Congruence (1) |
| | | | Others (1) |
| | | | Perpendicular Bisector (1) |
| | | | Right Triangle (4) |
| | | | Similarity (2) |
| | Solid Geometry (11) | Cylinder and Cone (4) | |
| | | Prism (2) | Oblique Prism (1) |
| | | | Right Prism (1) |
| | | Pyramid (2) | Regular Pyramid (2) |
| | | Section (2) | Straight Cut (2) |
| | | Sphere (1) | Tangent Plane (1) |
| Music (65) 0215 Music and performing arts | Chord Structures and Interval Diagrams (10) | Perfect Fifth and Major Third (3) | |
| | | Triads and Seventh Chords (7) | |
| | Key Signatures and Time Signatures (8) | Circle Of Fifths (3) | |
| | | Compound Triple Time (1) | |
| | | Key Signature (4) | |
| | Notes and Rests (47) | | |

| Level 1 | Level 2 | Level 3 | Level 4 |
|---|---|---|---|
| Physics (113)
0533 Physics
0714 Electronics and automation | Circuits and Electronics (36) | Circuit Diagram (25) | Curves In Circuit (2) |
| | | | DC and AC Circuit (19) |
| | | | Kirchhoff Law Application (2) |
| | | | Op Amp Circuit (2) |
| | | Component Symbols and Combinations (3) | Resistor (1) |
| | | | Transistor (2) |
| | | Signal and Electronics (8) | Filter (1) |
| | | | Spectrum Diagram (2) |
| | | | Waveform Diagram (5) |
| | Electromagnetism (15) | Electric and Magnetic Field (8) | |
| | | Electromagnetic Induction (7) | Ampere Force (1) |
| | | | Energy Loss and Hysteresis Loop (2) |
| | | | Lorentz Force (4) |
| | Mechanics (28) | Fluid Mechanics (23) | Gas Pressure (3) |
| | | | Liquid Pressure (15) |
| | | | Principle (5) |
| | | Kinematics (4) | |
| | | Newtonian Mechanics (1) | |
| | Optics and Waves (11) | Ray Diagram (9) | Interference and Diffraction (2) |
| | | | Lens Imaging (3) |
| | | | Reflection and Refraction (4) |
| | | Wave (2) | |
| | Quantum Mechanics (10) | Atomic Structure (2) | |
| | | Electromagnetic Wave (1) | |
| | | Photoelectric Effect (2) | |
| | | Potential Well (1) | |
| | | Spectrum (4) | |
| | Thermodynamics (13) | Intermolecular Forces (2) | |
| | | Molecular Speed Distribution (1) | |
| | | Phase Change (8) | |
| | | Pressure Volume Temperature Relationship (2) | |

# E   LIMITATIONS

While GenExam advances multidisciplinary T2I evaluation, its size (1000 samples) is still relatively small compared to multidisciplinary understanding benchmarks (*e.g.* 11.5K for MMMU) and may not cover all sub-disciplines and specified domains. In addition, the evaluation framework relies on MLLMs as judges – though enhanced by scoring points and ground truth, it still depends on the MLLMs' own disciplinary knowledge and understanding capability, potentially introducing bias or errors in assessing niche domain details.

# F PROMPTS

## F.1 AUTOMATIC FILTERING

You are an expert-level visual reasoning evaluator. You will be given an image. Your task is to assess a given image for its suitability as a basis for multidisciplinary (i.e. requires subject knowledge such as math, physics, chemistry, biology, etc.) image generation. From each of the perspectives below, you should assign a score with a confidence score and a short explanation.

## Perspectives

1. Text Richness Score (range: 0-10): If the image content is pure table/text without any other content (e.g. graphs, diagrams, plots, icons, geometry, etc.), it is 0. If the image does not contain any text, it is 10. For other cases, the score is based on the complexity of the image content.

2. Image Domain (range: 0-1): If the image is a natural image (i.e., real image) / pathological image (also including body scans, MRI, CT scans, and X-rays), it is 0, otherwise it is 1.

3. Image Complexity (range: 1-10): Whether the image is too complex to draw (consider the number and complexity of components,objects, text, etc.), very complex is 1, very simple is 10.

4. Subject Knowledge (range: 1-10): Whether the model needs **subject knowledge and reasoning** to generate this image (note: the prompt is not a complete description of the image, it is related to subject knowledge), no need is 1, need a lot is 10.

## Instructions

For each domain, provide:

- **Score**: An integer, based on the range defined above.

- **Confidence Score**: An integer between 1 and 5.

- **Short Explanation**: 1-2 sentences justifying the score, referencing the image content and its alignment with the domain definition. Make sure your explanation is concise.

## Output Format

Respond STRICTLY in the following format:

{{
    "Text Richness Score": {{
        "score": 0,
        "confidence": 0,
        "explanation": ""
    }},
    "Image Domain": {{
        "score": 0,
        "confidence": 0,
        "explanation": ""
    }},
    "Image Complexity": {{
        "score": 0,
        "confidence": 0,
        "explanation": ""
    }},
    "Subject Knowledge": {{
        "score": 0,
        "confidence": 0,
        "explanation": ""
    }},
}}

Make sure your response follows the format strictly and can be parsed by Python json.loads. Do not include any other explanation.

## F.2 ANNOTATING

You are an expert-level prompt generator for multidisciplinary text-to-image generation. You are given an image, a related conversation between a user and a model, and its corresponding taxonomy.

Your job is to provide a **prompt (like an academic question)** for multidisciplinary text-to-image generation corresponding to this image. You will also generate some **scoring points** when evaluating whether some model generates the correct image corresponding to this prompt.

## Instructions

### Prompt

1. The prompt should **NOT** be a complete description of the image. It should be in a style similar to an **academic question** that **requires some subject knowledge and reasoning** to generate the image. If a model does not have such subject knowledge, it is expected to fail to generate the correct image.

2. The prompt should be precise and concise (less than 200 words) and should be in English.

3. The provided conversation and text in the image can sometimes help you to generate the prompt, but you should consider them carefully since they may contain some irrelevant information.

## Scoring Points

1. The scoring points are in the form of a list of questions and their corresponding scores, answered by "Yes" or "No" only. They are designed so that by answering the questions, we can evaluate whether the model can generate the correct image corresponding to the prompt. It is expected that the answers are all "Yes" if the model generates the correct image.

2. The number of scoring points should be no more than 20. If the image is simple and requires little subject knowledge and reasoning, the number of scoring points can be small.

3. The questions should be based on the prompt instead of the image, i.e. since the model is required to generate the image based on the prompt, the questions should not focus on information or components that are in the ground truth image but not in the prompt.

4. The questions should not be too general. It should provide **details** rather than "Is xxx correct?", e.g. the structure of a molecule (bonds, position of atoms, etc.), characteristics that a curve should satisfy, etc. It should not be too complex as you should break it down into several sub-questions.

5. The score of each scoring point should be a number between 0 and 1, indicating the proportion of this question in the total score of the image.**The sum of all scores should be 1**.

6. Typically, the score of the most basic question (e.g. checking the overall structure of the image) should be small, such as 0.1 or 0.2. For complex questions, questions requiring subject knowledge, or questions about details, the score should be large.

## Examples

1. Prompt: "Generate a planar molecular structure diagram of benzene ($C_6H_6$), showing its conjugated double bond structure, and distinguish carbon atoms and hydrogen atoms with different colors.", Scoring points: [{{"question": "Is the number of carbon atoms 6?", "score": 0.2}}, {{"question": "Is the number of hydrogen atoms 6?", "score": 0.2}}, {{"question": "Is the number of double bonds 3?", "score": 0.2}}, {{"question": "Is the number of single bonds 3?", "score": 0.2}}, {{"question": "Are carbon atoms and hydrogen atoms in different colors?", "score": 0.2}}]

2. Prompt: "Generate the graph of the function $y = e^x$.", Scoring points: [{{"question": "Does the image shows a graph of a function?", "score": 0.1}}, {{"question": "Does the curve pass through the point (0, 1)?", "score": 0.2}}, {{"question": "Does the curve asymptotically approach the x-axis when x approaches negative infinity?", "score": 0.2}}, {{"question": "Does the curve tends to positive infinity when x approaches positive infinity?", "score": 0.2}}, {{"question": "Is the curve smooth and continuous?", "score": 0.1}}, {{"question": "Is the slope always positive and increasing when x increases?", "score": 0.2}}]

3. Prompt: "Generate a schematic diagram of an animal cell structure. Label cell membrane, cytoplasm, nucleus, and mitochondria.", Scoring points: [{{"question": "Does the image contain a cell structure?", "score": 0.1}}, {{"question": "Does the cell membrane label correspond to the correct position in the cell?", "score": 0.2}}, {{"question": "Does the cytoplasm label correspond to the correct position in the cell?", "score": 0.2}}, {{"question": "Does the nucleus label correspond to the correct position in the cell?", "score": 0.2}}, {{"question": "Does the mitochondria label correspond to the correct position in the cell?", "score": 0.3}}]

## Output Format

Respond STRICTLY in the following format:

{{
    "prompt": "",
    "scoring_points": [
        {{"question": "question1", "score": 0.1}},
        ...
    ],
}}

Make sure your response follows the format strictly and can be parsed by Python json.loads. Do not include any other explanation.

## Taxonomy of this image

{taxonomy}

## Conversation

{conversations}

## F.3 EVALUATION

You are an expert-level evaluator for multidisciplinary text-to-image generation. You will be given an input prompt for text-to-image generation, an image generated by a model (the first image), a ground truth image for reference (the second image), and several scoring point questions and their corresponding scores. Your task is to evaluate the generated image by answering the scoring point questions and evaluating the image from some global perspectives.

## Instructions

1. Remember that the first image is the generated image from a model, and the second image is the ground truth image. You should **only evaluate the generated image**, while the ground truth image can be used for reference.

2. First, give a detailed description of all the components in the generated image.

3. For each scoring point, you should use the ground truth image as **reference information** to make more accurate judgments. You should generate a **detailed** reasoning **step by step**. It should first analyze the question, what subject knowledge it requires and what information you need to refer to from the ground truth image, then analyze whether the generated image satisfies the scoring point based on the scoring point, prompt and reference information. Finally, answer 1 if the generated image fully satisfies the scoring point, otherwise answer 0. For example, if a scoring point is "Does the cell membrane label correspond to the correct position in the cell?", you can refer to the ground truth image to check the correct position of the cell membrane label as an example, by **explicitly analyzing the ground truth image** in your reasoning.

4. You should also evaluate the generated image from some global perspectives provided below. For each perspective, first provide a **detailed step-by-step** reasoning, e.g. recognize all the text labels for spelling, then give a score in the defined range.

## Global Perspectives

1. Spelling (range: 0-2): The spelling of the text in the image, including the notations and equations. You should first recognize the text in the image in the reasoning, then check the spelling of the text. Specifically:

- 0: There are critical errors in spelling, notations or equations which significantly hinders the understanding of the image.

- 1: There are some errors in spelling, notations or equations that somehow hinder the understanding of key information in the image.

- 2: All or almost all the spelling, notations and equations are correct. Tiny errors like commas, capital letters, etc. are allowed.

2. Readability (range: 0-2): The readability of the image. Each component in the image should be clearly readable and identifiable. All text labels and marks should in the right place and are not overlapped or occluded by other elements. If the image is geometry or diagram, there should not be unlabeled points or lines or duplicated labels. If the image is a plot or chart, the axis should be labeled and the ticks should be carefully checked. You should first identify the components, labels, and marks in the image in the reasoning, then check their readability. Specifically:

- 0: There are critical readability issues which significantly hinders the understanding of the image e.g. some components are impossible to distinguish, or some labels are fully overlapped or occluded, or many key information are not labeled.

- 1: There are some readability issues, e.g. some components are not clearly readable and identifiable, or some labels are overlapped or occluded, or some key information are not labeled.

- 2: The readability is perfect or almost perfect. Components are clearly readable and identifiable and necessary labels and marks are present. Tiny errors are allowed.

3. Logical Consistency (range: 0-2): The logical consistency of the image. Check the correctness of all the marks, text, musical notes, etc. If the image is geometry, check the correctness of each marked angles, lengths, coordinates, etc. If the image is a plot or chart, check the correctness of each data point, the axis, the ticks, the legend, etc. You should first identify the components, labels, and marks in the image in the reasoning, then check their logical consistency. Specifically:

- 0: There are critical logical consistency issues which significantly hinders the understanding of the image, e.g. some key marks are not correct, some angles/lengths/data points are significantly inconsistent with the text label, etc.

- 1: There are some logical consistency issues that somehow hinder the understanding of key information in the image, e.g. some marks are not correct, some angles/lengths/data points are inconsistent with the text label, etc.

- 2: The logical consistency is perfect or almost perfect. Marks, text, etc. are correct and consistent. Tiny errors are allowed.

## Output Format

Respond STRICTLY in the following format:

{{
    "description": "...",
    "answers": [
        {{
            "reasoning": "...",
            "answer": 1
        }},
        ...
    ],
    "global_evaluation": {{
        "Spelling": {{
            "reasoning": "...",
            "score": 0,

```
      }},
      "Readability": {{
        "reasoning": "...",
        "score": 0,
      }},
      "Logical Consistency": {{
        "reasoning": "...",
        "score": 0,
      }},
    }}
}}
```

Make sure your response follows the format strictly and can be parsed by Python json.loads. Do not include any other explanation.

## Prompt

{prompt}

## Scoring Points

{scoring_points}

## G MORE VISUALIZATION

More visualization of generated images are given in Figs. 11, 12 and 13.

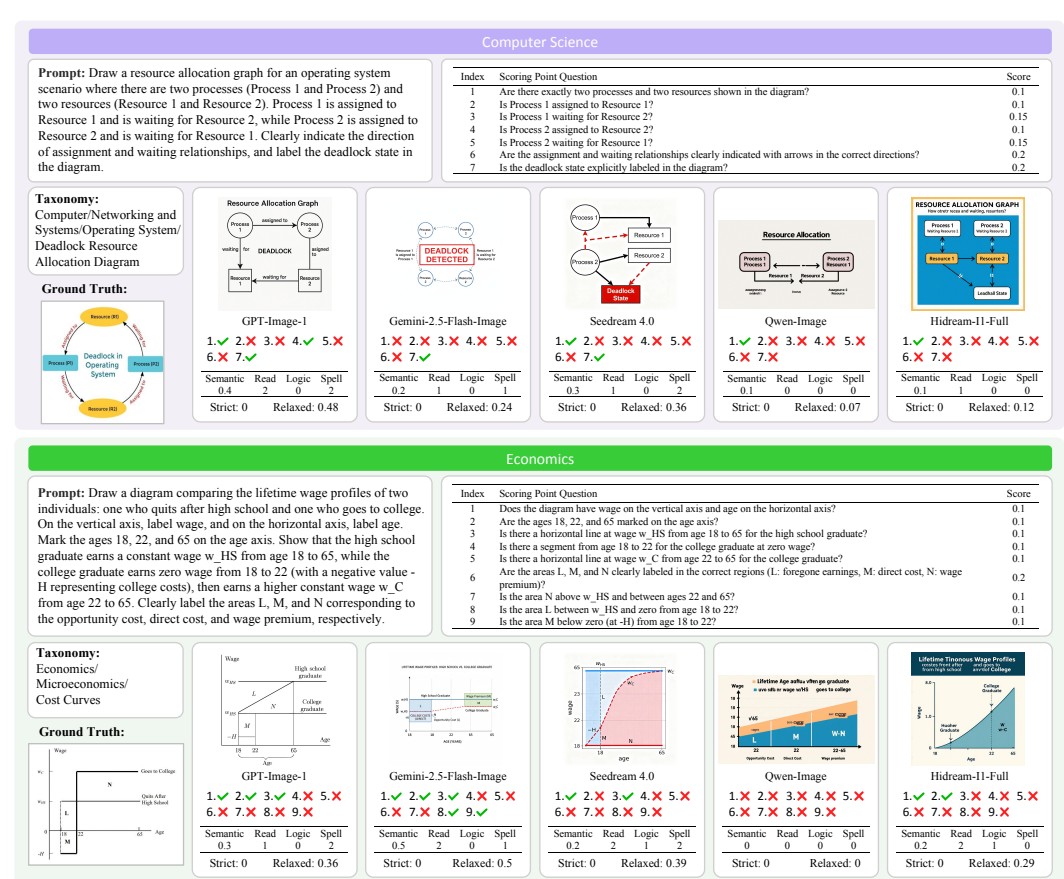

Figure 11: **Visualization of generated images.**

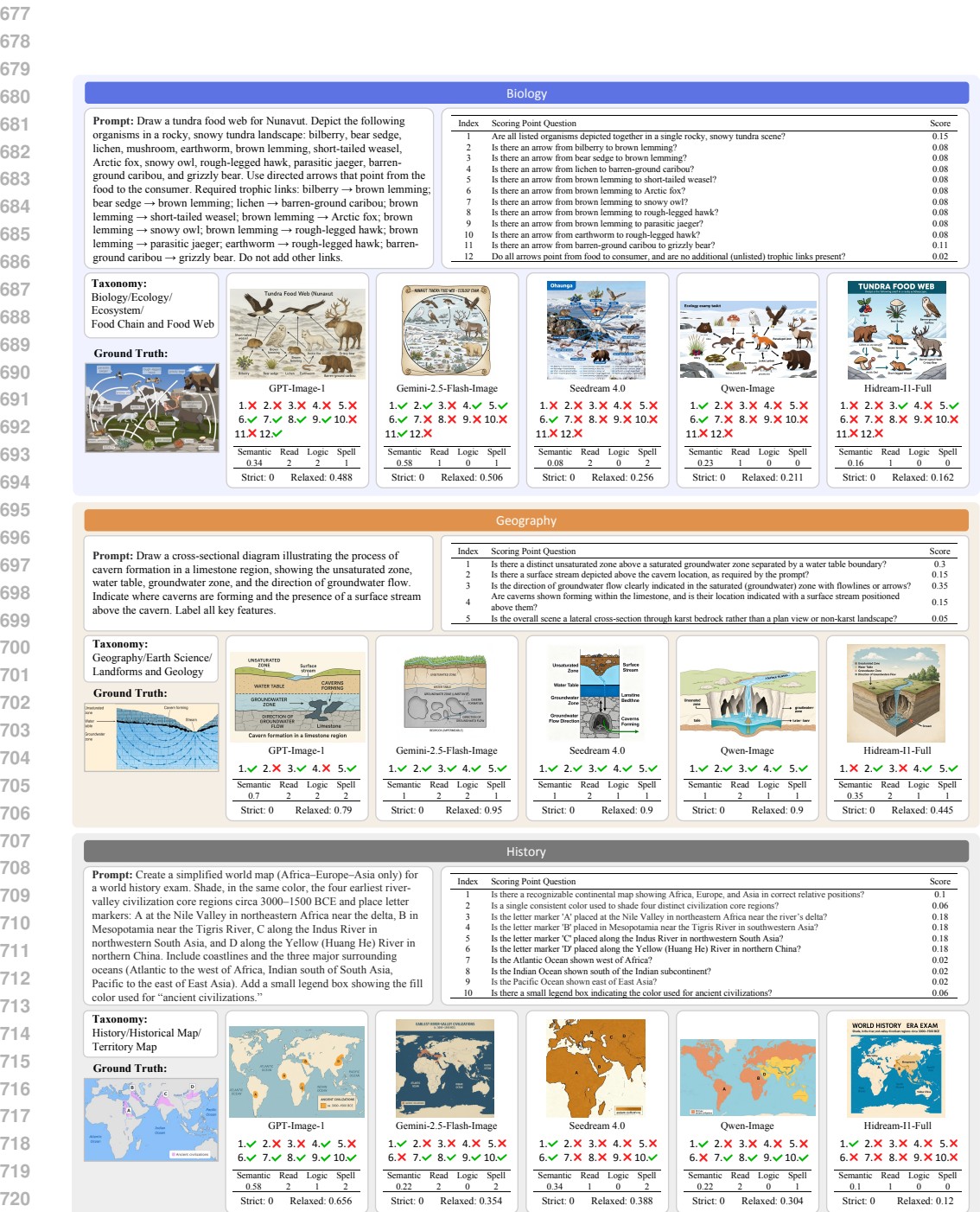

Figure 12: **Visualization of generated images.**

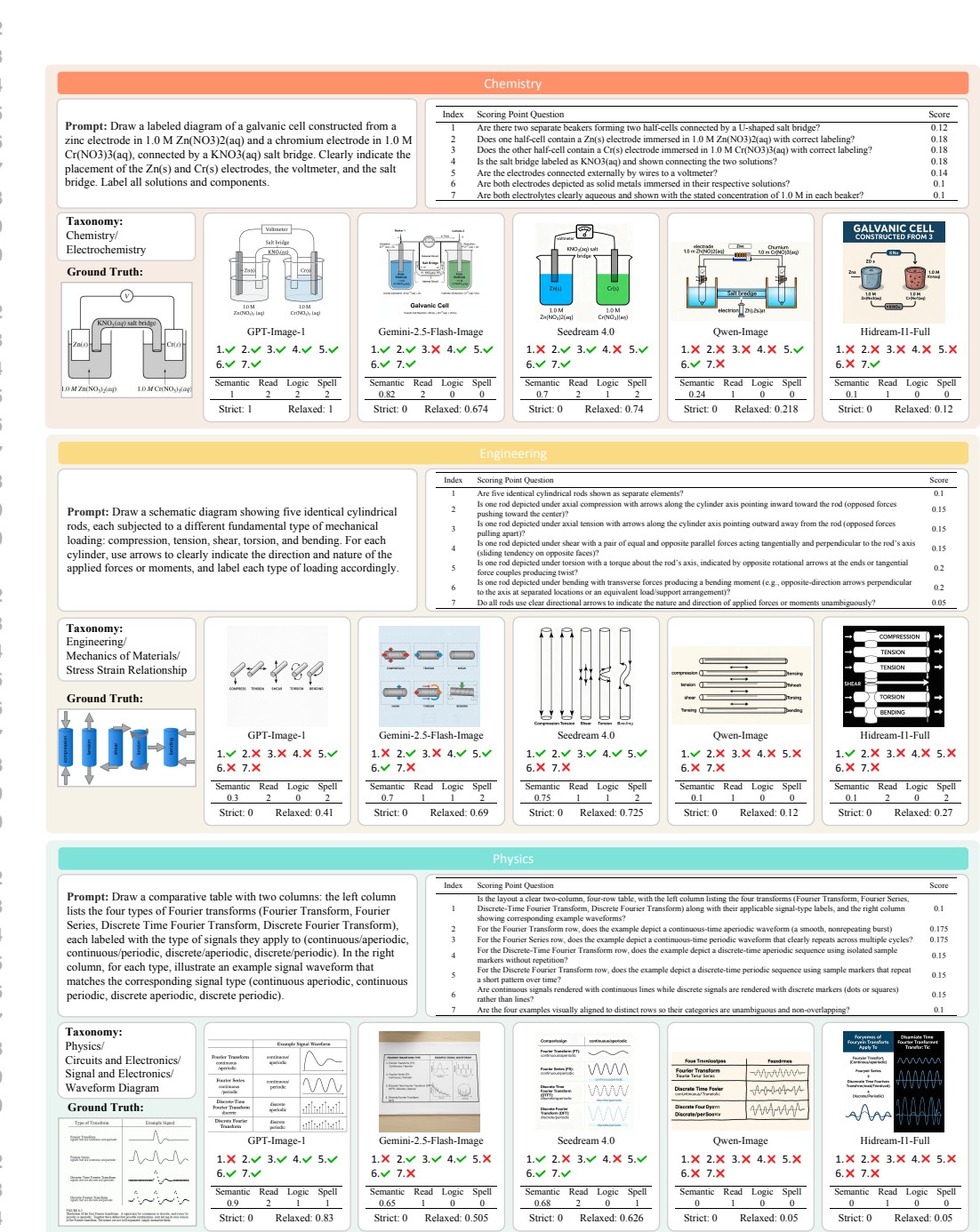

Figure 13: **Visualization of generated images.**

