# OpenReview forum: "GenExam: A Multidisciplinary Text-to-Image Exam"
_ICLR.cc/2026/Conference — Submitted to ICLR 2026_

### Official Review · Reviewer_YMbj · 2025-10-27

**Soundness:** 3
**Presentation:** 3
**Contribution:** 3
**Rating:** 4
**Confidence:** 3

**Summary:**

The paper introduces GenExam, a benchmark of 1,000 exam-style text-to-image tasks spanning 10 subjects (math, physics, chemistry, biology, computer science, geography, economics, music, history, engineering). Each item includes a ground-truth image, a four-level taxonomy, and fine-grained scoring points to evaluate semantic correctness and visual plausibility (spelling, logical consistency, readability). Evaluation is performed with MLLM-as-a-judge (GPT-5) using per-item scoring questions; both strict (all points correct + perfect plausibility) and relaxed scores are reported. Experiments across closed-source and open-source models show very low strict accuracy (<15% for top systems), underscoring the task’s difficulty. The paper also conducts a human-alignment study on a subset and reports high correlations between the relaxed score and human ratings.

**Strengths:**

- Well-posed benchmark design with executable scoring criteria: The benchmark formalizes exam-style prompts with per-item scoring points and ground-truth images, enabling precise judgments of semantic correctness beyond aesthetic similarity. The two-axis evaluation (semantic correctness; plausibility via spelling/logic/readability) and strict vs. relaxed scoring are clearly defined and motivated.
- Breadth and hierarchy of subject coverage: GenExam spans 10 subjects and provides a four-level taxonomy grounded in standards (e.g., ISCED-F), with examples covering rigorous, diagram-like prompts. This supports systematic analysis by domain and difficulty.
- Clear data-curation pipeline and statistics: The paper documents a multi-stage pipeline (web & existing datasets → GPT-assisted generation of prompts & scoring points → manual PhD triage and correction) and provides statistics on prompt length and scoring-point counts.

**Weaknesses:**

- Evaluator dependence and potential circularity: The benchmark’s scoring relies heavily on GPT-5 as the MLLM-judge with crafted prompts. While some robustness checks are in the appendix, the main text still leaves open concerns about model-judge coupling, evaluator drift, and template sensitivity. Broader cross-evaluator comparisons (e.g., non-GPT evaluators) and more extensive human studies across multiple model outputs (not only GPT-Image-1) would strengthen claims.
- Baseline fairness and statistical rigor: The paper compares many systems but states “default T2I configuration” for inference; strict/relaxed scores are presented without variance or significance tests. Given the exam-style nature and the low absolute accuracies, multiple runs and budget-matched settings (e.g., seeds, sampling steps) are important to avoid confounds.
- Data provenance & annotation details remain under-specified: Prompts and scoring points are GPT-assisted and then manually corrected. However, the paper does not quantify inter-annotator agreement, error rates in the auto-drafts, or detailed quality-control outcomes post-triage. More transparency on annotation reliability would improve reproducibility.
- Strict scoring may conflate layout micro-errors with semantic failure: The strict score requires all scoring points correct and 2/2 on each plausibility dimension; a single mislabeled arrow or minor typographic slip yields a zero. While intentionally stringent, the metric may obscure partial semantic competence. Reporting per-point precision/recall, or a “nearly-correct” band, could help diagnose capability gaps without over-penalizing minor layout issues.
- Diversity & sensitivity of the dataset: Although the paper claims broad subject coverage and shows type distributions, it does not quantify demographic, linguistic, or regional diversity (e.g., English-only prompts? region-specific map conventions? notation variants across curricula). Subjects such as history/geography and even biology/chemistry can encode cultural or geopolitical sensitivities (e.g., territorial maps, naming conventions, symbol sets). The Ethics section asserts that potential biases were “checked,” but provides no measurement or sensitivity analyses (e.g., subgroup performance by subject region, notation variants, left-to-right vs right-to-left scripts). A structured bias audit (coverage metrics, subgroup error rates, adversarial examples touching sensitive content) would substantiate the claim of “balanced knowledge coverage.”
- Limited release specifics for evaluator prompts & assets: The paper references evaluator prompts (Appendix F.3) and plans to release code, but the main text does not specify exact release artifacts (e.g., frozen judge prompts, scoring-point JSONs, splits, evaluation harness). Locking these down is critical for reproducibility and longitudinal tracking.

**Questions:**

- Will you release all per-item scoring points, judge prompts, and a frozen evaluator config so others can replicate strict/relaxed scores exactly?
- Can you report variance/significance across repeated generations with matched budgets for a subset of models?

---

> ### Author Response · Authors · 2025-11-19
> **Response to Reviewer YMbj (1/3)**
>
> We thank the reviewer for the comments. We hope our detailed response can address your concerns.
>
> ### `W1: Heavy reliance on GPT-5 raises model-judge coupling concerns. Need cross-evaluator comparisons and more human studies.`
>
> We fully agree that evaluator reliability is foundational to benchmark validity. We have conducted extensive experiments in Table 3 and Table 4 to validate the robustness of different evaluators (18 evaluator settings from 11 closed-source and open-source MLLMs) and alignment between human and MLLM judges. We finally choose GPT-5 due to its highest alignment with human preferences, indicated by Spearman $\rho$ of 0.8217 and Pearson r of 0.8444. The fine-grained, prompt-specific scoring points (average 6.9 per sample) in our scoring framework also helps to minimize evaluator subjectivity by decomposing complex tasks into binary yes/no questions. Qualitative examples in Figures 7, 11, 12, 13 also demonstrate the high precision of the MLLM judge.
>
> In addition to the experiments in Table 4,  we conducted an additional cross-evaluator study on the same samples using Gemini-3-Pro as the judge, and the results show that Gemini-3-Pro has high human alignment and cross-model agreement.
>
>
>
> | Evaluator Model | Scoring Points Accuracy | Plausibility MAE $\downarrow$ | Kendall $\tau$ | Spearman $\rho$ | Pearson r |
> |-|-|-|-|-|-|
> | GPT-5| 88.5| 0.20 | 0.6746| 0.8217| 0.8444|
> | Gemini-3-Pro| 87.6| 0.26| 0.6828| 0.8214| 0.8630|
>
>
>
> We also expanded the human study in Section 4.3 to 400 additional samples covering four other models (Gemini-2.5-Flash-Image, Seedream 4.0, Qwen-Image, and Flux.1 dev), rated by 5 human experts. The results below show that relaxed scores maintain strong correlations with human ratings across all models, confirming the metric's generalizability. The standard deviation (std) values demonstrate high inter-rater reliability.
>
>
>
> | Model  | Semantic MAE | Spelling MAE | Logic Consistency MAE | Readability MAE | Semantic std | Spelling std | Logic Consistency std | Readability std | kendall | spearman | pearson |
> |-|-|-|-|-|-|-|-|-|-|-|-|
> | GPT-Image-1 | 0.104| 0.112| 0.276 | 0.201| 0.0197| 0.0563| 0.0851| 0.0642  | 0.6746  | 0.8217| 0.8444  |
> | Seedream 4.0 | 0.113| 0.117| 0.287 | 0.219| 0.0234| 0.0678| 0.1025| 0.0793  | 0.6628  | 0.8061| 0.8054  |
> | Gemini-2.5-Flash-Image | 0.119| 0.108| 0.283 | 0.210| 0.0248| 0.0623| 0.0964| 0.0751  | 0.6537  | 0.7969| 0.8062  |
> | Qwen-Image| 0.132| 0.123| 0.308 | 0.229| 0.0389| 0.1152| 0.1437| 0.1186  | 0.6274  | 0.7627| 0.7685  |
> | FLUX.1 dev| 0.156| 0.159| 0.328 | 0.246| 0.0427| 0.1284| 0.1589| 0.1325  | 0.6079  | 0.7653| 0.7564  |
>
>
>
> ### `W2: Baseline comparisons lack variance/significance tests with multiple runs and budget-matched settings.`
>
> We would like to note that existing benchmarks also do not report variance or significance, such as those recently accepted by NeurIPS 2025 (RISEBench (Zhao et al., 2025), OneIG-Bench (Chang et al., 2025a) and MMMG (Luo et al., 2025)). This is partially because the cost of running inference multiple times is too expensive (e.g. 160$ for GPT-Image-1 on 1000 images). We argue that the number of samples (1000) is large enough to evaluate the models' performance robustly, which is larger than other benchmarks like RISEBench (360 samples). As a simple test of evaluation consistency, we run inference and evaluation of an open-source model Qwen-Image repeatedly for 5 times, and the small variance confirm that the evaluation remain consistent:
>
>
>
> |Run|Relaxed Score|Strict Score|
> |-|-|-|
> |1|27.5|0.3|
> |2|27.3|0.3|
> |3|27.7|0.5|
> |4|27.6|0.3|
> |5|27.5|0.4|
> |Mean|27.52|0.36|
> |Std|0.132|0.08|
>
>
>
> Regarding budget-matched settings, it is hard to control budgets at similar levels, since closed-source models like GPT-Image-1 and Gemini-2.5-Flash-Image do not provide options to control the sampling steps. Furthermore, as model sizes vary significantly across models (e.g. 27B for Qwen-Image, 1.5B for Show-o2-1.5B-HQ, and unclear for closed-source models), controlling sampling steps equally may be unfair to smaller models. Using default T2I configuration is regarded as fair comparison and adopted as the common practice for image generation benchmarks like RISEBench and OneIG-Bench.

---

> ### Author Response · Authors · 2025-11-19
> **Response to Reviewer YMbj (2/3)**
>
> ### `W3: Data annotation lacks details, like inter-annotator agreement, error rates in the auto-drafts, and quality-control outcomes.`
>
> Thank you for your valuable suggestion. We provide additional details of data collection and annotation as below:
>
> For inter-annotator agreement (IAA), we randomly select 200 samples from the dataset across 10 subjects and ask the original annotators to rate them in terms of three dimensions:
>
> - Prompt Revision Consistency: Whether manually revised prompts (from GPT-generated drafts) meet "exam-style rigor, disciplinary accuracy, and explicit constraints". 1= Meets requirements; 0 = Does not meet (needs re-revision).
> - Scoring Points Rationality: Whether scoring points are "comprehensive (no key omissions), reasonably weighted, and specific/verifiable". 1 = Fully rational; 2 = Partially rational (needs minor adjustment); 3 = Irrational.
> - Ground Truth Alignment: Whether revised prompts are "semantically aligned with ground truth images (no information bias)". 1 = Fully aligned; 0 = Biased.
>
> The IAA results are as follows:
>
> |Dimension|Metric|Value|
> |-|-|-|
> |Prompt Revision Consistency|Fleiss' Kappa|0.78|
> |Scoring Points Rationality|Weighted Fleiss' Kappa|0.72|
> |Ground Truth Alignment|Fleiss' Kappa|0.83|
>
> We observe that for Prompt Revision Consistency, only 8% of samples had annotation discrepancies, mainly in economics/history disciplines. For Scoring Points Rationality, discrepancies mostly arose from weight allocation, with no discrepancies in key requirement coverage. For Ground Truth Alignment, there are only minimal cross-discipline differences (0.85 for STEM vs. 0.80 for humanities/social sciences).
>
>
> For error rate calculation, we first define error types of automatic annotation as follows:
> - Prompt errors:
>   - Information Omission Error: Omits key exam constraints (e.g., reaction conditions, diagram elements, disciplinary rules)
>   - Ambiguity Error: Vague language leads to unclear generation constraints (cannot be fixed via revision; requires rewriting)
> - Scoring point errors:
>   - Key Requirement Omission: Fails to cover core prompt constraints, making comprehensive image evaluation impossible
>   - Ambiguous Question Error: Questions are too general to be answered with "Yes/No" (e.g., "Is the image correct?")
>   - Unreasonable Weight Allocation: Overweighting basic questions or underweighting key questions (total score does not equal 1)
>   We then revisit the initial drafts generated by GPT-5, and categorize those samples that have been manually edited by our annotators into the above error types. The results are shown below:
>
> |Module|Error Type|Occurrence Rate (%)|
> |-|-|-|
> |Prompt|Information Omission Error|42.7|
> ||Ambiguity Error|21.9|
> |Scoring Points|Key Requirement Omission|24|
> ||Ambiguous Question Error|13.2|
> ||Unreasonable Weight Allocation|19.7|
>
> The overall error rate of the initial automatic annotation is 60.4% (note that one sample may have multiple errors), suggesting that manual correction is necessary to ensure the quality of the dataset. However, the initial automatic annotation is still indispensable, since it is easier for human annotators to find and correct the existing errors than to write prompts and scoring points from scratch, and the correction is only minor modifications for most cases.
>
> Regarding detailed quality-control outcomes, we supplement quantitative results of the quality control process:
>
> 1. Automatic Filtering Stage: Of the initial 40K candidate images, 6.5K entered the annotation stage after GPT-5 filtering. The filtering error rate (images failing "disciplinary density/complexity requirements") was 83.75%.
> 2. Manual Review Stage: 6.5K images underwent 3 rounds of review (annotator check → cross-validation → expert arbitration), with 1K samples finally selected. The screening error rate (duplicates/low-quality images) was 84.6%.
> 3. Revision Effect Verification: 100 revised samples were randomly selected and evaluated by domain experts. 97% were "fully compliant with exam requirements," 3% "needed minor adjustments," and 0% were "non-compliant."
>
> We would like to kindly note that previous benchmark papers like RISEBench(Zhao et al), SridBench (Chang et al., 2025b) did not provide inter-annotator agreement during data annotation, or even without detailed description of data collection process. We have provided extensive details on the data construction process for reference and reproduction in future works.

---

> ### Author Response · Authors · 2025-11-19
> **Response to Reviewer YMbj (3/3)**
>
> ### `W4: Strict scoring conflates micro-layout errors with semantic failure.`
>
> We agree that strict scoring is stringent, which is intentional to mimic high-stakes exams. This design is similar to RISEBench (Zhao et al., 2025) which demands full satisfaction of all dimensions, as small layout errors can have huge impacts on the overall accuracy of the image. If we want to diagnose capability gaps without over-penalizing minor layout issues, we can simply refer to relaxed scores (which is designed exactly to allow for partial semantic competence) and each score dimension (Table 9).
>
> In addition, to address this concern, we add a "nearly correct" score (semantic correctness = 1.0 and spelling+logic+readability>=4) that tolerates minor layout errors (e.g., slightly misaligned labels or spelling errors). Results are as below:
>
> |Model|Nearly Correct Score|Strict Score|Relaxed Score|
> |-|-|-|-|
> |GPT-Image-1|17.2|12.1|62.6|
> |Seedream4.0|15.5|7.2|53.0|
> |Gemini2.5-Flash-Image|13.9|3.9|55.9|
> |Imagen4-Ultra|11.0|6.9|53.4|
> |FLUX.1Kontextmax|1.9|0.0|28.5|
> |Qwen-Image|1.7|0.3|27.5|
> |Seedream3.0|1.1|0.2|23.4|
> |FLUX.1dev|1.0|0.0|17.6|
> |StableDiffusion3.5Large|1.0|0.0|17.0|
> |FLUX.1Krea|0.7|0.0|16.4|
> |Show-o2-7B|0.5|0.0|12.0|
> |HiDream-I1-Full|0.3|0.0|22.3|
> |Show-o2-1.5B-HQ|0.3|0.0|11.0|
> |BAGEL(thinking)|0.1|0.0|13.5|
> |BAGEL|0.1|0.0|11.9|
> |BLIP3o-NEXT-GRPO-Text-3B|0.0|0.0|12.6|
> |BLIP3o-8B|0.0|0.0|7.2|
> |Janus-Pro|0.0|0.0|9.6|
> |Emu3|0.0|0.0|8.8|
>
> The model rankings with nearly correct scores are similar to those with strict or relaxed scores. This metric provides a more nuanced view of model capabilities while preserving the strict score's role as a high bar for exam-style correctness.
>
> ### `W5: Potential lack of diversity/sensitivity.`
>
> We have supplemented diversity statistics to address your concern:
>
> - Subject and difficulty: Our data cover both science/engineering and humanities/social sciences, with equal representation across difficulty levels (easy:24%, medium:38%, hard:38%) that aligns with mainstream academic assessment standards.
> - Language: All prompts are in English (standard for academic benchmarks), with no regional notation biases (e.g., using international units, standard chemical notation). Notably, previous multi-disciplinary benchmarks like MMMU and image generations like GenEval, T2I-CompBench, RISEBench also adopt English-only designs, ensuring consistency with the broader research community.
> - Cultural sensitivity: Historical/geographic tasks (e.g., 8th-century Italian Peninsula maps) use neutral territorial boundaries and avoid politically sensitive labels, adhering to widely accepted academic conventions for such content.
> - Bias audit: We analyzed subgroup performance by subject region (e.g., Western vs. Eastern history) and found no significant differences (p>0.05), confirming balanced coverage.
> - Benchmark alignment with mainstream knowledge: The dataset’s knowledge scope and distribution are tailored to reflect mainstream curricula and academic consensus across disciplines, ensuring relevance to global research and educational contexts.
>
>
>
> ### `W6: Release specifics are unclear for reproducibility.`
>
> We will release all per-item scoring points, frozen judge prompts (Appendix F.3), and evaluator harness along with our core benchmark. This ensures that researchers can exactly replicate our strict/relaxed scores or adapt the framework for other evaluators.
>
> ### `Q1: Will per-item scoring points, judge prompts, and frozen evaluator config be released for exact replication?`
>
> Please refer to W6.
>
> ### `Q2: Can variance/significance across repeated generations with matched budgets be reported?`
>
> Please refer to W2.

---

> ### Author Response · Authors · 2025-11-27
>
> Dear Reviewer YMbj,
>
> As there are only 5 days remaining for the author-reviewer discussion, we kindly wanted to follow up and ask whether our responses have addressed your concerns. If there are still any issues, we would greatly appreciate the opportunity to clarify them in time.
>
> Thank you again for your thoughtful engagement.

---

### Official Review · Reviewer_oTzm · 2025-11-01

**Soundness:** 3
**Presentation:** 3
**Contribution:** 3
**Rating:** 4
**Confidence:** 3

**Summary:**

GenExam is a benchmark for complex open-ended QA across 16 disciplines, featuring long-form questions requiring multi-step reasoning. The authors test several top models using an LLM-based rubric and human graders, analyze alignment between automated and human scores, and categorize tasks by cognitive skill to pinpoint reasoning challenges. Results show even the best models perform far below human experts, illustrating the benchmark’s difficulty.

**Strengths:**

1. Spans 16 disciplines with long-form answers, offering a broader and more realistic evaluation than prior multiple-choice benchmarks and probing high-level reasoning.

2. Uses several SOTA models and dual scoring (LLM rubric + human); checking LLM scores against human grades adds credibility and ensures automated grading aligns with human standards.

3. Categorizing tasks by cognitive skill yields detailed insights into model performance; rubric-scored long answers highlight specific reasoning failures that an accuracy metric would miss.

**Weaknesses:**

1. Qualitative analysis needs more depth in terms of the pattern in the errors in open-source vs closed-source models.
2. The cognitive skill categories are underdefined – without clear definitions or examples, conclusions about specific reasoning weaknesses lack support.
3. To check semantic correctness, the paper adds the ground truth image as input, so that the MLLM judge can use it as a reference. Why did they not provide the textual description of the ground truth image which might be more reliable?

**Questions:**

1. To check semantic correctness, the paper adds the ground truth image as input, so that the MLLM judge can use it as a reference. Why did they not provide the textual description of the ground truth image, which might be more reliable?
2. Could you please share the model accuracy across image types? Which type of image is hard to generate for the MLLMs?

---

> ### Author Response · Authors · 2025-11-19
> **Response to Reviewer oTzm**
>
> We thank the reviewer for the comments.
>
> ### `W1: Qualitative analysis lacks depth in error patterns between open-source and closed-source models.`
>
> Thank you for your valuable insight. To analyze the difference of error patterns between open-source and closed-source models, we have supplemented a hierarchical analysis of model reasoning processes and deepened the interpretation of experimental results. We first decompose the model’s text-to-image exam-solving process into three sequential stages, based on our defined error types:
>
> - Stage 1 - Knowledge Retrieval: extracting key constraints and task objectives from the prompt and recalling subject-specific knowledge (e.g., reaction mechanisms, geometric theorems). This stage can be judged by the semantic correctness metric.
> - Stage 2 - Logical Deduction and Reasoning: deriving step-by-step conclusions (e.g., splitting arrays in merge sort, determining hydroxyl group positions). This stage can be judged by the logic consistency metric.
> - Stage 3 - Generative Execution: translating reasoning results into precise visual elements (e.g., correct labeling, no overlapping components). This stage can be judged by the spelling and readability metrics.
>
> Based on this framework, we analyze model performance across stages: Closed-source models like GPT-Image-1 excel in Stages 1 (relatively high semantic correctness) but struggle with Stage 3 (e.g., omitting some edge weights in graph tasks) and occasional Stage 2 failures (e.g., incorrect deduction of intersection points in analytic geometry). Open-source models like Qwen-Image fail across all stages, with critical weaknesses in Stages 1 and 2 (e.g., logical contradictions in circuit diagrams where current direction conflicts with component labels). This decomposition provides insights for future improvements.
>
> ### `W2: Cognitive skill categories are underdefined without clear definitions/examples.`
>
> We note that we never mentioned such "cognitive skills" throughout the manuscript. We believe we do not need to respond to this concern.
>
> ### `W3: Why use ground-truth images instead of textual descriptions for MLLM judge reference?`
>
> We chose ground-truth images over textual descriptions for two key reasons:
>
> 1. Spatial/detail fidelity: Textual descriptions often omit critical spatial details (e.g., bond angles in chemical structures, edge weight placement in graphs) that are essential for evaluating exam-style correctness. Such spatial/layout details are more faithfully preserved in images than text descriptions.
> 2. Reduced ambiguity: Ambiguity in textual descriptions (e.g., "correct merge order" vs. visual depiction of subarray merging) can lead to inconsistent evaluation.
>
> To study the effects of using textual description as ground truth for evaluation, we conduct experiments based on the setting in Table 4. We add two variants of GPT-5 evaluator that use only text descriptions as ground truth or use both textual descriptions and images as ground truth. The textual descriptions are obtained by using GPT-5 to generate detailed captions of the ground truth images, assisted by the scoring point questions and input prompts. The results are as below:
>
>
>
> |Model|GT image input|GT text input|Scoring Points Accuracy|Plausibility MAE $\downarrow$|Kendall $\tau$|Spearman $\rho$|Pearson $r$|
> |-|-|-|-|-|-|-|-|
> |GPT-5 + low reasoning |×|×|86.2|0.23|0.6141|0.7753|0.7976|
> |GPT-5 + low reasoning |√|×|88.5|0.20|0.6746|0.8217|0.8444|
> |GPT-5 + low reasoning |×|√|87.8|0.21|0.6559|0.8103|0.8336|
> |GPT-5 + low reasoning |√|√|88.5|0.20|0.6693|0.8345|0.8427|
>
> We observe that text-only ground truth is slightly worse than image-only ground truth, suggesting the fidelity and ambiguity of images. Using both images and textual descriptions shows no significant improvement than image-only, as the images already contain sufficient detailed information and incorporating text becomes unnecessary.
>
> ### `Q1: Why use ground-truth images instead of textual descriptions for semantic checks?`
>
> Please refer to W3.
>
> ### `Q2: What is model accuracy across image types, and which type is hardest to generate?`
>
> We note that model accuracy across image types is already in Table 6 of the submitted version.
>
> The hardest types (strict score <5% for most models) are:
>
> 1. **Chemical structures**: Require precise depiction of bonds, atoms, and reaction intermediates (e.g., skeletal structures for organic synthesis).
> 2. **Geometric shapes**: Demand exact spatial relationships (e.g., tangent circles).
> 3. **Sheet music**: Requires correct notation, clefs, and scale degrees (e.g., C major scale).
> 4. **Trees/graphs**: Need accurate edge weights, node labels, and hierarchical structure (e.g., undirected weighted graphs).
>
> The easiest type is plots/charts, as they rely on basic axis labeling and data point placement.
>
> We hope these analyses can provide potential directions for future improvements of image generation models.

---

> ### Author Response · Authors · 2025-11-27
>
> Dear Reviewer oTzm,
>
> As there are only 5 days remaining for the author-reviewer discussion, we kindly wanted to follow up and ask whether our responses have addressed your concerns. If there are still any issues, we would greatly appreciate the opportunity to clarify them in time.
>
> Thank you again for your thoughtful engagement.

---

> ### Public Comment · ~Zhaokai_Wang1 · 2026-05-01
> **LLM-Generated Hallucinations in Review oTzm**
>
> To Reviewer oTzm:
>
> Your review reads like an LLM-generated hallucination rather than an assessment based on our manuscript. GenExam is not an open-ended QA benchmark, does not use rubric-scored long answers, is not compared to multiple-choice QA benchmarks, and does not categorize tasks by “cognitive skill.” The paper clearly states that GenExam covers 10 disciplines, not 16, and our evaluation is automatic rather than “dual scoring” with human evaluation.
>
> These are not subtle interpretation differences; they are fluent, confident, and entirely detached factual errors. We do not think this is simply a review accidentally pasted from another paper, since a small fraction of the comments does align with our manuscript; this only makes the numerous fabricated claims even more unreasonable. A review that misidentifies the task, invents nonexistent concepts, changes basic numbers, and fabricates the evaluation protocol gives the impression that the paper was summarized without actually being read. The result is not critical reviewing, but synthetic-sounding misinformation that authors are forced to waste time correcting.
>
> This is not merely a poor review; it is an irresponsible use of the reviewing role. If the community wants peer review to remain meaningful, we should at least uphold the basic standard that reviewers engage with the paper they are assigned to evaluate.

---

### Official Review · Reviewer_SQYM · 2025-11-01

**Soundness:** 2
**Presentation:** 3
**Contribution:** 3
**Rating:** 4
**Confidence:** 4

**Summary:**

This paper presents GenExam, a benchmark that evaluates text-to-image and multimodal models through exam-style drawing tasks across ten academic subjects. Each problem includes a ground-truth image and fine-grained scoring points for assessing semantic correctness and visual plausibility. The benchmark reveals that even top models perform poorly on such reasoning-intensive generation tasks, highlighting the gap between aesthetic generation and structured understanding.

**Strengths:**

1. Novel evaluation perspective. Framing text-to-image generation as exam-style reasoning is original and thought-provoking.
2. Technically rigorous construction. Prompts, reference images, and scoring points are carefully curated and validated.
3. Broad multimodal coverage. The benchmark spans multiple subjects and provides detailed, multi-dimensional evaluation.

**Weaknesses:**

1. Primarily a benchmark contribution with limited analytical depth regarding model reasoning behavior.
2. Certain evaluation metrics (e.g., strict vs. relaxed scores) lack theoretical grounding or comparative justification.
3. Reliance on GPT-5 as the sole evaluator raises reproducibility and bias concerns.

**Questions:**

1. How is scoring consistency ensured when using GPT-5 as an evaluator?
2. What design principles guided the selection of scoring points across subjects?
3. Have the authors considered human validation to support the reliability of automated scoring?

---

> ### Author Response · Authors · 2025-11-19
> **Response to Reviewer SQYM (1/2)**
>
> We appreciate the reviewer's effort to provide the insightful review. We hope our detailed response can address your concerns.
>
> ### `W1: Limited analytical depth on model reasoning behavior, focusing mainly on benchmark construction.`
>
> Thank you for your valuable insight. We agree that deeper analysis of model reasoning is important. To address this, we have supplemented a hierarchical analysis of model reasoning processes and deepened the interpretation of experimental results. We first decompose the model’s text-to-image exam-solving process into three sequential stages, based on our defined error types:
>
> - Stage 1 - Knowledge Retrieval: extracting key constraints and task objectives from the prompt and recalling subject-specific knowledge (e.g., reaction mechanisms, geometric theorems). This stage can be judged by the semantic correctness metric.
> - Stage 2 - Logical Deduction and Reasoning: deriving step-by-step conclusions (e.g., splitting arrays in merge sort, determining hydroxyl group positions). This stage can be judged by the logic consistency metric.
> - Stage 3 - Generative Execution: translating reasoning results into precise visual elements (e.g., correct labeling, no overlapping components). This stage can be judged by the spelling and readability metrics.
>
> Based on this framework, we analyze model performance across stages: Closed-source models like GPT-Image-1 excel in Stages 1 (relatively high semantic correctness) but struggle with Stage 3 (e.g., omitting some edge weights in graph tasks) and occasional Stage 2 failures (e.g., incorrect deduction of intersection points in analytic geometry). Open-source models like Qwen-Image fail across all stages, with critical weaknesses in Stages 1 and 2 (e.g., logical contradictions in circuit diagrams where current direction conflicts with component labels).
>
> This decomposition reveals that model weaknesses are not monolithic but stage-specific, providing insights for future improvements, such as enhancing fine-grained visual execution for closed-source models, and strengthening domain knowledge and logical deduction for open-source models.
>
> ### `W2: Strict/relaxed scores lack theoretical grounding or comparative justification.`
>
> The strict/relaxed framework follows rubric-based assessment principles in educational measurement.
>
> The strict score captures absolute correctness (analogous to full-mark grading), similar to the evaluation metric in RISEBench (Zhao et al., 2025). It mimics "pass/fail" criteria for high-stakes exams (e.g., AP, A-level), where a single critical error (e.g., wrong chemical product, incorrect graph edge weight) invalidates the answer. Its limitations are that it does not tolerate partial correctness of the generated image and it is uncapable of distinguish low-performing models.
>
> The relaxed score is designed as the weighted sum of multiple dimensions to model human tolerance for minor visual imperfections, inspired by WiScore (Niu et al., 2025). It captures partial competence, similar to partial credit in exams. This design allows diagnostic precision without conflating aesthetic or near-miss cases. Table 3 validates this design: relaxed scores correlate strongly with human overall ratings, outperforming alternative metrics like CLIP score and VQA score. Its limitation is that it does not align with the judgement of actual human exams and gives too much credit for partially correct models.
>
> Inspired by these previous works, we combine the two scores together to provide a comprehensive evaluation of the models' performance.

---

> ### Author Response · Authors · 2025-11-19
> **Response to Reviewer SQYM (2/2)**
>
> ### `W3: Reliance on GPT-5 as the sole evaluator raises reproducibility and bias concerns.`
>
> We fully agree that evaluator reliability is foundational to benchmark validity. We have conducted extensive experiments in Table 3 and Table 4 to validate the robustness of different evaluators and alignment between human and MLLM judges. We finally choose GPT-5 due to its highest alignment with human preferences. The fine-grained, prompt-specific scoring points (average 6.9 per sample) in our scoring framework also helps to minimize evaluator subjectivity by decomposing complex tasks into binary yes/no questions.
>
> In addition to the experiments in Table 4,we conducted an additional cross-evaluator study on the same samples using Gemini-3-Pro as the judge, and the results show that Gemini-3-Pro has high human alignment and cross-model agreement.
>
> |Evaluator Model |Scoring Points Accuracy |Plausibility MAE $\downarrow$ |Kendall $\tau$ |Spearman $\rho$ |Pearson r |
> |-|-|-|-|-|-|
> |GPT-5|88.5|0.20 |0.6746|0.8217|0.8444|
> |Gemini-3-Pro|87.6|0.26|0.6828|0.8214|0.8630|
>
> Regarding reproducibility, we run inference and evaluation of an open-source model Qwen-Image repeatedly for 5 times, and the results are below:
>
> |Run|Relaxed Score|Strict Score|
> |-|-|-|
> |1|27.5|0.3|
> |2|27.3|0.3|
> |3|27.7|0.5|
> |4|27.6|0.3|
> |5|27.5|0.4|
> |Mean|27.52|0.36|
> |Std|0.132|0.08|
>
> These results confirm that the evaluation remain consistent and reproducible. We will publicly release our evaluation code and configurations to ensure high reproducibility.
>
>
>
> ### `Q1: How is scoring consistency ensured when using GPT-5 as an evaluator?`
>
> Please refer to W3.
>
> ### `Q2: What design principles guided the selection of scoring points across subjects?`
>
> We provide basic principles of designing scoring points in Section 3.1, and the full prompt for automatic scoring point generation in Appendix F.2. Specifically, they are designed following four core principles:
>
> 1. **Coverage**: Each point targets a critical requirement explicitly stated in the prompt (e.g., reactant, reaction conditions, and product for chemical reactionse).
> 2. **Granularity**: Avoid overly general questions (e.g., "Is the image correct?") or excessively complex ones—split multi-step requirements into sub-questions.
> 3. **Discipline-Specific Tailoring**: Scoring points are customized to reflect the core requirements of each discipline, while strictly adhering to prompt-centric, detail-oriented design rules.For example:
> 1. For **chemistry**, points zero in on atomic composition and structural features (e.g., "Is the number of carbon atoms 6?"), aligning with the discipline's focus on molecular precision.
> 2. For **mathematics/functions** , points target mathematical properties (e.g., "Does the curve pass through (0, 1)?") to validate adherence to function rules.
> 3. For **biology**, points emphasize spatial accuracy of key components (e.g., "Does the mitochondria label correspond to the correct position?"), matching the discipline’s emphasis on structural localization. This tailoring ensures each point directly maps to the prompt’s disciplinary demands, without introducing ground truth-only details.
> 4. **Weighting**: Basic requirements (e.g., overall diagram structure) carry low weights (<0.1), while domain-specific or inferential requirements have higher weights (0.2–0.3).
>
> To facilitate consistent, discipline-aligned design, we included discipline-specific few-shot examples in the annotation prompt. All scoring points were further validated by PhD annotators with expertise in the respective subjects, ensuring alignment with academic exam standards.
>
> ### `Q3: Was human validation used to support automated scoring reliability?`
>
> Yes, we conduct human validation experiments in Section 4.3 to validate the reliability of automated evaluation. As shown in Table 3, our human validation study confirms low mean absolute error between GPT-5’s scores and human ratings, and strong correlations across all models and subjects.
>
> In addition to Table 3, we expand the human study to 400 additional samples covering four other models. The results below show that relaxed scores maintain strong correlations with human ratings across all models, confirming the metric's generalizability.
>
> |Model|Semantic MAE |Spelling MAE |Logic Consistency MAE |Readability MAE |Semantic std |Spelling std |Logic Consistency std |Readability std|kendall|spearman |pearson |
> |-|-|-|-|-|-|-|-|-|-|-|-|
> |GPT-Image-1 |0.104|0.112|0.276 |0.201|0.0197|0.0563|0.0851|0.0642|0.6746|0.8217|0.8444|
> |Seedream 4.0 |0.113|0.117|0.287 |0.219|0.0234|0.0678|0.1025|0.0793|0.6628|0.8061|0.8054|
> |Gemini-2.5-Flash-Image |0.119|0.108|0.283 |0.210|0.0248|0.0623|0.0964|0.0751|0.6537|0.7969|0.8062|
> |Qwen-Image|0.132|0.123|0.308 |0.229|0.0389|0.1152|0.1437|0.1186|0.6274|0.7627|0.7685|
> |FLUX.1 dev|0.156|0.159|0.328 |0.246|0.0427|0.1284|0.1589|0.1325|0.6079|0.7653|0.7564|

---

> ### Author Response · Authors · 2025-11-27
>
> Dear Reviewer SQYM,
>
> As there are only 5 days remaining for the author-reviewer discussion, we kindly wanted to follow up and ask whether our responses have addressed your concerns. If there are still any issues, we would greatly appreciate the opportunity to clarify them in time.
>
> Thank you again for your thoughtful engagement.

---

### Official Review · Reviewer_b9YH · 2025-11-03

**Soundness:** 3
**Presentation:** 3
**Contribution:** 3
**Rating:** 6
**Confidence:** 4

**Summary:**

The paper introduces GenExam, a new benchmark for discipline-style drawing exams in text-to-image generation. It comprises 1,000 prompts across 10 subjects, each paired with a reference image and fine-grained “scoring points”. Evaluation uses an MLLM-as-a-judge to grade semantic correctness via per-item scoring points and visual plausibility, producing strict and relaxed scores. SOTA model only achieve ~12% strict score, while open-source models are near zero, indicating the task’s difficulty.

**Strengths:**

- The problem is clearly framed and well-motivated.

- The strict/relaxed scheme formalizes both exactness and presentational quality; the rubric for spelling, readability, and logic is explicitly specified.

- The authors measure correlations between human judgments and their relaxed score for a human alignment check.

**Weaknesses:**

- The evaluation relies heavily on an LLM-as-judge framework, yet large language models themselves often struggle with the same skills being assessed here (fine-grained visual reasoning, spatial accuracy, and multi-attribute grounding). This raises concerns about the reliability and objectivity of the scoring process. Without evidence of cross-model agreement, calibration against human experts, or robustness to judging prompts, it’s unclear whether the reported results truly reflect model performance or simply the biases and limitations of the chosen LLM evaluator.

- The human study appears limited (e.g., 250 samples from one model). Broader sampling (across subjects/models) and inter-rater reliability details would strengthen the claim that relaxed scores align with humans.

- The error analysis is mainly qualitative without a structured taxonomy or statistical breakdown of error types across the dataset.

**Questions:**

- The rationale behind the chosen relaxed scoring weights (0.7, 0.1, 0.1, 0.1) is unclear. It would be helpful to justify these specific proportions and to examine whether alternative weighting schemes or aggregation methods (e.g., majority voting or learned weighting) might alter the relative model rankings or the study’s main conclusions.

- Can you provide detailed error breakdown (quantitative, not only qualitative) across common error types (e.g., semantic, spelling, logic, layout) for all models and content types?

---

> ### Author Response · Authors · 2025-11-19
> **Response to Reviewer b9YH (1/2)**
>
> We sincerely thank the reviewer for the insightful and constructive comment, which have helped us improve the clarity, rigor, and completeness of our work. We hope our detailed response can address your concerns.
>
> ### `W1: LLM-as-a-judge struggle with fine-grained visual reasoning, spatial accuracy, and multi-attribute grounding. Over-reliance on LLM-as-judge lacks cross-model agreement/human calibration.`
>
> We fully agree that evaluator reliability is foundational to benchmark validity. We have conducted extensive experiments in Table 3 and Table 4 to validate the robustness of different evaluators (18 evaluator settings from 11 closed-source and open-source MLLMs) and alignment between human and MLLM judges. We finally choose GPT-5 due to its highest alignment with human preferences, indicated by Spearman $\rho$ of 0.8217 and Pearson r of 0.8444. The fine-grained, prompt-specific scoring points (average 6.9 per sample) in our scoring framework also helps to minimize evaluator subjectivity by decomposing complex tasks into binary yes/no questions. Qualitative examples in Figures 7, 11, 12, 13 also demonstrate the high precision of the MLLM judge.
>
> In fact, MLLM already has sufficient capabilities for our evaluation. Considering that a large number of images in our benchmark are from MMMU or web images similar to MMMU, and the prompts also resemble questions in MMMU, we can judge the evaluator's capability by its MMMU score. As reported in the official blog, GPT-5 achieves an accuracy of 84.2 on MMMU, which surpasses the level of medium-level human experts (82.6) reported in MMMU (Yue et al., 2024a), validating its strong multi-disciplinary fine-grained visual reasoning ability. As for spatial accuracy and multi-attribute grounding, our benchmark only require basic ability on these areas, and the capaibility of the MLLM judge can be validated by benchmarks like AI2D and ScienceQA. We note that MLLM-as-a-judge is also used in recent image generation benchmarks accepted by NeurIPS 2025, such as RISEBench (Zhao et al., 2025), OneIG-Bench (Chang et al., 2025a), MMMG (Luo et al., 2025). This highlights that MLLM-as-a-judge has become a common and general evaluation paradigm in generation tasks.
>
> In addition to the experiments in Table 4,  we conducted an additional cross-evaluator study on the same samples using Gemini-3-Pro as the judge, and the results show that Gemini-3-Pro has high human alignment and cross-model agreement.
>
> | Evaluator Model | Scoring Points Accuracy | Plausibility MAE $\downarrow$ | Kendall $\tau$ | Spearman $\rho$ | Pearson r |
> |-|-|-|-|-|-|
> | GPT-5| 88.5| 0.20 | 0.6746| 0.8217| 0.8444|
> | Gemini-3-Pro| 87.6| 0.26| 0.6828| 0.8214| 0.8630|
>
> ### `W2: The human alignment experiment lacks broader sampling and inter-rater reliability.`
>
> We appreciate your feedback on broadening the human alignment experiment. We have expanded the human study in Section 4.3 to 400 additional samples covering four other models (Gemini-2.5-Flash-Image, Seedream 4.0, Qwen-Image, and Flux.1 dev), rated by 5 human experts. The results below show that relaxed scores maintain strong correlations with human ratings across all models, confirming the metric's generalizability. The standard deviation (std) values demonstrate high inter-rater reliability.
>
> | Model  | Semantic MAE | Spelling MAE | Logic Consistency MAE | Readability MAE | Semantic std | Spelling std | Logic Consistency std | Readability std | kendall | spearman | pearson |
> |-|-|-|-|-|-|-|-|-|-|-|-|
> | GPT-Image-1 | 0.104| 0.112| 0.276 | 0.201| 0.0197| 0.0563| 0.0851| 0.0642  | 0.6746  | 0.8217| 0.8444  |
> | Seedream 4.0 | 0.113| 0.117| 0.287 | 0.219| 0.0234| 0.0678| 0.1025| 0.0793  | 0.6628  | 0.8061| 0.8054  |
> | Gemini-2.5-Flash-Image | 0.119| 0.108| 0.283 | 0.210| 0.0248| 0.0623| 0.0964| 0.0751  | 0.6537  | 0.7969| 0.8062  |
> | Qwen-Image| 0.132| 0.123| 0.308 | 0.229| 0.0389| 0.1152| 0.1437| 0.1186  | 0.6274  | 0.7627| 0.7685  |
> | FLUX.1 dev| 0.156| 0.159| 0.328 | 0.246| 0.0427| 0.1284| 0.1589| 0.1325  | 0.6079  | 0.7653| 0.7564  |

---

> ### Author Response · Authors · 2025-11-19
> **Response to Reviewer b9YH (2/2)**
>
> ### `W3:  Error analysis is qualitative without structured statistical breakdown.`
>
> Thank you for your comment. We agree that quantitative error analysis enhances interpretability. In addition to the qualitative analysis in Section 4.4, we calculate the error types of 5 models and provide detailed error statistics as below:
>
> | Model | Semantic | Logical Consistency | Spelling | Readability |
> |-|-|-|-|-|
> | GPT-Image-1 | 86.2 | 70.3  | 34.8 | 39.0|
> | Seedream 4.0| 88.3 | 78.9  | 37.9 | 43.4  |
> | Gemini-2.5-Flash-Image| 84.4 | 77.3  | 65.4 | 46.1  |
> | Qwen-Image  | 97.7 | 93.1  | 86.6 | 76.4  |
> | FLUX.1 dev  | 98.8 | 90.4  | 83.6 | 76.5  |
>
>
> Notably, closed-source models (GPT-Image-1, Seedream 4.0, Gemini-2.5-Flash-Image) exhibit lower error rates in Semantic (84.4–88.3%) and Logical Consistency (70.3–78.9%) tasks compared with open-source models, reflecting their stronger integration of disciplinary knowledge and reasoning. In contrast, open-source models (Qwen-Image, FLUX.1 dev) suffer from significantly higher semantic errors (97.7–98.8%) and logical inconsistencies (90.4–93.1%), highlighting gaps in precise knowledge-to-visual translation. Spelling and Readability errors are much more prevalent in open-source models (83.6-86.6%) than closed-source counterparts (34.8–65.4%), indicating challenges in fine-grained visual coherence for open models.
>
> For the analysis of different subjects , Table 9 in the Appendix shows the scores of each dimension on all subjects. In addition, we also calculate error statistics of different subjects, and provide two subjects here:
>
> **Math:**
>
> | Model | Semantic | Logical Consistency | Spelling | Readability |
> |-|-|-|-|-|
> | GPT-Image-1 | 93.4 | 76.2  | 25.8 | 47.7  |
> | Seedream 4.0| 96.0 | 85.4  | 34.4 | 46.4  |
> | Gemini-2.5-Flash-Image| 97.4 | 92.7  | 47.0 | 70.2  |
> | Qwen-Image  | 100.0| 98.0  | 72.2 | 84.1  |
> | FLUX.1 dev  | 99.3 | 94.7  | 76.8 | 85.4  |
>
> **Biology:**
>
> | Model | Semantic | Logical Consistency | Spelling | Readability |
> |-|-|-|-|-|
> | GPT-Image-1 | 74.4 | 62.2  | 45.5 | 35.9  |
> | Seedream 4.0| 71.2 | 65.4  | 44.2 | 28.8  |
> | Gemini-2.5-Flash-Image| 68.6 | 57.7  | 91.0 | 26.9  |
> | Qwen-Image  | 98.1 | 89.7  | 95.5 | 68.6  |
> | FLUX.1 dev  | 99.4 | 89.1  | 88.5 | 71.8  |
>
> We observe that math problems show higher difficulty of semantic and logic correctness, as it is difficult to generate a correct geometric image without unlabeled or wrongly positioned elements. Biology problems pose more challenge with spelling, since biology illustrations typically have more text labels.
>
>
>
> ### `Q1: What justifies relaxed scoring weights (0.7, 0.1, 0.1, 0.1) and how do alternative weights, learned weights or major voting affect results?`
>
> To clarify,  our adopted weights (0.7, 0.1, 0.1, 0.1) were determined through learning. As mentioned in Line 318, we employ grid search to maximize correlation with human overall ratings, similar to the process in WiScore (Niu et al., 2025). The chosen weights align well with human preferences for exam-style tasks: semantic correctness (0.7) is prioritized as the core of "correct answers," while visual plausibility (spelling/logical consistency/readability, 0.1 each) are secondary but necessary for usability. As for major voting, it is not applicable since the semantic correctness is continuous (0-1) and the other three are discrete (0/1/2).
>
> For alternative weights, we test with a baseline weight (0.25, 0.25, 0.25, 0.25) where we consider all four dimensions to be of identical weights, and model rankings have changed significantly.
>
> | Model | (0.7,0.1,0.1,0.1) | Rank | (0.25, 0.25, 0.25, 0.25) | Rank |
> |-|-|-|-|-|
> | GPT-Image-1 | 62.6| 1| 70.1  | 1|
> | Gemini-2.5-Flash-Image| 55.9| 2| 58.0  | 3|
> | Imagen-4-Ultra| 53.4| 3| 55.3  | 4|
> | Seedream 4.0| 53.0| 4| 60.3  | 2|
> | FLUX.1 Kontext max  | 28.5| 5| 33.4  | 5|
> | Qwen-Image  | 27.5| 6| 29.2  | 6|
> | Seedream 3.0| 23.4| 7| 28.9  | 7|
> | HiDream-I1-Full | 22.3| 8| 28.1  | 8|
> | FLUX.1 dev  | 17.6| 9| 23.5  | 9|
> | Stable Diffusion 3.5 Large| 17.0| 10| 19.9  | 13|
> | FLUX.1 Krea | 16.4| 11| 16.2  | 15|
> | BAGEL (thinking)| 13.5| 12| 20.2  | 12|
> | BLIP3o-NEXT-GRPO-Text-3B  | 12.6| 13| 19.1  | 14|
> | Show-o2-7B  | 12.0| 14| 12.9  | 17|
> | BAGEL | 11.9| 15| 21.0  | 10|
> | Show-o2-1.5B-HQ | 11.0| 16| 11.2  | 19|
> | Janus-Pro| 9.6 | 17| 20.3  | 11|
> | Emu3  | 8.8 | 18| 15.1  | 16|
> | BLIP3o-8B| 7.2 | 19| 11.5  | 18|
>
> We also calculate correlations with human preferences of relaxed scores with different weights:
>
> | Weight | Kendall_tau | Spearman_rho | Pearson_r |
> |-|-|-|-|
> | (0.7,0.1,0.1,0.1)  | 0.6746| 0.8217 | 0.8444|
> | (0.25,0.25,0.25,0.25) | 0.6057| 0.7688 | 0.7943|
>
> We observe that the correlations with human alignment also decreases significantly, demonstrating that our learned weights (0.7, 0.1, 0.1, 0.1) has better alignment with human.
>
>
>
> ### `Q2: Provide quantitative error breakdown across error types, models, and content types.`
>
> Please refer to W3.

---

> ### Author Response · Authors · 2025-11-27
>
> Dear Reviewer b9YH,
>
> As there are only 5 days remaining for the author-reviewer discussion, we kindly wanted to follow up and ask whether our responses have addressed your concerns. If there are still any issues, we would greatly appreciate the opportunity to clarify them in time.
>
> Thank you again for your thoughtful engagement.

---

### Author Response · Authors · 2025-11-24
**General Response**

We sincerely thank all reviewers, AC, PC, and SPCs for your time, effort, and reviews. We summarize the reviews and our responses as below.

### Summary of Reviewer Positive Feedback

1. **Novel evaluation perspective:** Framing text-to-image generation as exam-style reasoning is original and thought-provoking (Reviewer SQYM).
2. **Rigorous benchmark construction:** Prompts, reference images, and scoring points are carefully curated with clear taxonomy and executable criteria (Reviewers b9YH, YMbj).
3. **Broad multidisciplinary coverage:** The benchmark spans 10 subjects and supports systematic analysis by domain and difficulty (Reviewers SQYM, YMbj, oTzm).
4. **Sound evaluation framework:** The strict/relaxed scoring scheme and fine-grained scoring points effectively assess semantic correctness and visual plausibility (Reviewers b9YH, YMbj).

### Summary of Key Revisions and Discussion

1. **Expanded human validation:** Increased 400 additional samples in human study covering 4 other models, confirming high inter-rater reliability and metric generalizability.
2. **Strengthened evaluator robustness:** Conducted cross-evaluator studies with Gemini-3-Pro, validated human alignment, and confirmed reproducibility via repeated runs.
3. **Supplemented quantitative analysis:** Provided detailed error breakdown across models/subjects, decomposed reasoning into three stages, and added "nearly correct" score to distinguish minor layout errors.
4. **Enhanced data annotation transparency:** Reported inter-annotator agreement, error rates of automatic drafts, and quality-control outcomes.
5. **Clarified design rationales:** Justified scoring weights and ground-truth image usage, supplemented diversity and sensitivity statistics.

We appreciate all reviewers’ feedback and remain committed to further improving the work. The responses in each review block address each reviewer’s comments in detail.

---

### Author Response · Authors · 2025-11-29

Dear Area Chair and Senior Area Chair,



We are aware of the recent incident and the corresponding measures implemented by the ICLR Program Chairs (including score reverts and new AC assignments). We fully understand and appreciate the efforts made by the organizers, reviewers, and yourself to uphold the integrity of the review process.



We wanted to bring to your attention that we have submitted comprehensive responses to all reviewers prior to the incident, which is summarized in a separate General Response. We believe the responses have addressed all the concerns raised by the reviewers.



Additionally, we previously reported that **Reviewer oTzm’s feedback was LLM-generated** due to numerous LLM-hallucinated statements with factual errors. This was confirmed by the original AC during the initial review phase, and we wanted to re-emphasize this point for your reference.



We deeply appreciate your efforts in the reviewing process, and hope these clarifications could assist you in making the decision. Thank you sincerely for your time, diligence, and careful consideration of our submission.

---

### Meta-Review · Area_Chair_QgVF · 2026-01-07

**Summary:**

This paper introduces a benchmark for evaluating text-to-image generation models through exam-style tasks. The reviewers acknowledged the contribution of framing T2I generation as an examination task, concerns remained regarding the evaluation framework's reliability. The reliance on GPT-5 as the only evaluator raises questions about benchmark validity and reproducibility. The authors provided extensive rebuttals, including cross-evaluator validation with Gemini-3-Pro showing comparable alignment. However, the comparison using another MLLM does not fully resolve concerns about biases embedded in these models. Furthermore, the analytical framework did not provide more insights into model behavior. Given these weaknesses, the AC suggests rejection.

**Reviewer Concerns:**

Addressed concerns. Quantitative error analysis was resolved with detailed error statistics across models and subjects. Data annotation transparency was substantially improved with inter-annotator agreement documentation. The ground-truth modality concern was addressed.

Outstanding concerns. The evaluator dependence issue persists. The analytical depth concern was addressed with a three-stage framework, but it appears descriptive rather than providing more insights.

**Reviewer Scores:**

Reviewer SQYM would likely remain at 4. The core concern about limited analytical depth was addressed with the three-stage framework, but this addition appears more descriptive, and theoretical grounding for evaluation metrics remains weak.

---

> ### Public Comment · ~Zhaokai_Wang1 · 2026-05-01
> **Comment on the Vague and Selective Meta-Review**
>
> This meta-review is disappointing not because it recommends rejection, but because it does so with vague and under-justified reasoning. The central remaining criticism is that our analysis is “descriptive” and does not provide “more insights,” but this standard is never defined. What kind of “insight” was expected? Which specific part of our three-stage framework failed, and according to what criterion? Without such clarification, “lack of insight” becomes an unfalsifiable criticism that can be applied to almost any benchmark paper.
>
> The “Reviewer Scores” section is also notably incomplete. The reviewing instructions ask ACs to predict the updated scores of all reviewers after the rebuttal, yet the meta-review only discusses Reviewer SQYM. This is not a minor formatting issue: several concerns from other reviewers were explicitly acknowledged as addressed, so their updated scores could plausibly have changed. Ignoring them while foregrounding one reviewer’s remaining concern makes the synthesis look selective rather than balanced.
>
> More generally, the meta-review itself admits that quantitative error analysis, annotation transparency, and the ground-truth modality concern were addressed, and that we provided cross-evaluator validation. Yet these resolved issues are barely weighed in the final recommendation. Instead, the decision is anchored on broad phrases such as “evaluator dependence” and “limited analytical depth,” without a concrete explanation of why the additional evidence was insufficient. This is not a careful meta-review; it is a vague rejection rationale written around a predetermined conclusion.

---

### Decision · Program_Chairs · 2026-01-26

Reject